# Structure and functional properties of Norrin mimic Wnt for signalling with Frizzled4, Lrp5/6, and proteoglycan

**Tao-Hsin Chang, Fu-Lien Hsieh[†], Matthias Zebisch[‡], Karl Harlos, Jonathan Elegheert, E Yvonne Jones***

Division of Structural Biology, Wellcome Trust Centre for Human Genetics, University of Oxford, Oxford, United Kingdom

**Abstract** Wnt signalling regulates multiple processes including angiogenesis, inflammation, and tumorigenesis. Norrin (Norrie Disease Protein) is a cystine-knot like growth factor. Although unrelated to Wnt, Norrin activates the Wnt/β-catenin pathway. Signal complex formation involves Frizzled4 (Fz4), low-density lipoprotein receptor related protein 5/6 (Lrp5/6), Tetraspanin-12 and glycosaminoglycans (GAGs). Here, we report crystallographic and small-angle X-ray scattering analyses of Norrin in complex with Fz4 cysteine-rich domain ($Fz4_{CRD}$), of this complex bound with GAG analogues, and of unliganded Norrin and $Fz4_{CRD}$. Our structural, biophysical and cellular data, map Fz4 and putative Lrp5/6 binding sites to distinct patches on Norrin, and reveal a GAG binding site spanning Norrin and $Fz4_{CRD}$. These results explain numerous disease-associated mutations. Comparison with the *Xenopus* Wnt8–mouse $Fz8_{CRD}$ complex reveals Norrin mimics Wnt for Frizzled recognition. The production and characterization of wild-type and mutant Norrins reported here open new avenues for the development of therapeutics to combat abnormal Norrin/Wnt signalling.

*For correspondence: yvonne@
strubi.ox.ac.uk

Present address: [†]Department of
Biochemistry, University of
Oxford, Oxford, United
Kingdom; [‡]Evotec Ltd., Oxford,
United Kingdom

Competing interests: The
authors declare that no
competing interests exist.

Reviewing editor: John Kuriyan,
Howard Hughes Medical
Institute, University of California,
Berkeley, United States

## Introduction

Wnt morphogens, secreted cysteine-rich palmitoleoylated glycoproteins, play critical roles in cell-fate determination, tissue homeostasis and embryonic development (*Clevers and Nusse, 2012*; *Malinauskas and Jones, 2014*). Aberrant Wnt signalling leads to cancer, osteoporosis and degenerative illnesses (*Anastas and Moon, 2013*). Norrie Disease Protein (*NDP*) gene encodes Norrin (*Berger et al., 1992*; *Chen et al., 1992*), a secreted cystine-knot like growth factor, distinct from the lipid-modified Wnt (*Willert et al., 2003*). Norrin activates the canonical Wnt/β-catenin pathway by interaction with Wnt receptor Frizzled4 cysteine-rich domain ($Fz4_{CRD}$), and co-receptor low density lipoprotein receptor related protein 5/6 ectodomain ($Lrp5/6_{ECD}$), plus the auxiliary four-pass transmembrane protein Tetraspanin-12 (Tspan-12) and glycosaminoglycans (GAGs) of heparan sulfate proteoglycans (HSPGs) (*Xu et al., 2004*; *Junge et al., 2009*; *Ke et al., 2013*).

The Norrin mediated pathway maintains the blood-retina and blood-brain barriers (*Wang et al., 2012*) and regulates angiogenesis in the cochlea and uterus (*Rehm et al., 2002*; *Ye et al., 2011*) as well as neuroprotective effects on retinal neurons (*Ohlmann et al., 2010*; *Seitz et al., 2010*). Mutations in the *NDP* gene and the receptor genes, *FZ4*, *LRP5*, and *TSPAN-12*, have been identified for vitreoretinal diseases including Norrie Disease, Familial Exudative Vitreoretinopathy, and Coats' Disease (*Nikopoulos et al., 2010*; *Ye et al., 2010*; *Ohlmann and Tamm, 2012*). *NDP*, *FZ4*, *LRP5*, and *TSPAN-12* knock-out mice experiments further support the notion that dysfunctional Norrin signalling results in impaired retinal angiogenesis (*Richter et al., 1998*; *Kato et al., 2002*; *Robitaille et al., 2002*; *Xu et al., 2004*; *Junge et al., 2009*). Unlike Wnts which have promiscuous interactions with Fz receptors, Norrin specifically binds to $Fz4_{CRD}$, but not to the 14 other CRDs of Fz and secreted

**eLife digest** The cells within an animal need to be able to communicate with each other to coordinate many complex processes in the body, such as the formation of tissues and organs. One way in which the cells can communicate is through a pathway called Wnt signalling. Generally, one cell releases a protein called Wnt, which binds to a receptor protein called Frizzled that sits on the surface of the same or another cell. This activates a series of events in the cells that can change the activity of particular genes. Wnt signalling has many roles in animals, and defects in it can contribute to cancer and other devastating diseases.

Another protein called Norrin can also activate Wnt signalling by binding to Frizzled and another receptor protein called Lrp5/6. This group or 'complex' also includes molecules called glycosaminoglycans. In humans, mutations in the gene that encodes Norrin can cause a disease in which blood vessels in the eye fail to form correctly, which can result in blindness. However, it is not clear how Norrin activates Wnt signalling.

Chang et al. developed a method to produce large quantities of Norrin protein to allow them to study the structure of the protein. Then, a technique called X-ray crystallography was used to reveal the three-dimensional structure of Norrin when it is bound to Frizzled. The model reveals that a pair of Norrin proteins form a complex with two Frizzled proteins and highlights particular areas of the Norrin protein that interact with Frizzled. Molecules of glycosaminoglycan bind to a site in the complex that spans both Norrin and Frizzled. The model also predicts that other areas of the Norrin protein may be involved in binding Lrp5/6.

Chang et al. compared the model to the structure of a Wnt protein bound to Frizzled, which revealed that Norrin and Wnt show some fundamental similarities in the way they bind to Frizzled. These findings move us closer to defining the essential features of the protein complexes that modify Wnt signalling, and may aid the development of new therapies for diseases that affect the development of the eye.

Frizzled-related protein (sFRP) family members (*Hsieh et al., 1999*; *Smallwood et al., 2007*). Similar to Wnt, Norrin (1) binds to Lrp5/6$_{ECD}$ (*Ke et al., 2013*); (2) interacts with HSPGs and shows limited spatial diffusion (*Perez-Vilar and Hill, 1997*; *Xu et al., 2004*; *Smallwood et al., 2007*; *Ohlmann et al., 2010*). As well as being a potential target for therapeutic interventions, an understanding of Norrin mediated signalling will also provide insights into the fundamental features required to trigger canonical Wnt/β-catenin signalling.

Structural analyses of the extracellular components and interactions mediating Norrin signalling were considered to be challenging because of the difficulties of generating recombinant Norrin (*Perez-Vilar and Hill, 1997*; *Shastry and Trese, 2003*; *Ohlmann et al., 2010*). *Ke et al. (2013)* reported a refolding method (from *Escherichia coli* inclusion bodies) to produce active recombinant Norrin fused with a N-terminal maltose binding protein (MBP-Norrin), an advance that enabled them to determine the crystal structure of MBP-Norrin. Here, we develop an efficient mammalian cell expression method to produce active untagged recombinant Norrin and detail the structural and functional properties of this potential therapeutic agent. Our crystallographic and solution studies further reveal that dimeric Norrin forms a complex with two copies of monomeric Fz4$_{CRD}$. Our molecular level analysis of the Norrin–Fz4$_{CRD}$ complex bound with GAG analogue, in combination with structure-guided biophysical and cell-based studies, defines the basis for ligand recognition. Structural comparison with the *Xenopus* Wnt8 in complex with mouse Fz8$_{CRD}$ (*Janda et al., 2012*) shows that Norrin uses its β-strands to mimic a finger-like loop in Wnt for binding to the Fz receptor CRD. Finally, we note that engineered Norrin mutants resulting from our analyses may be of use as agents for blocking Wnt receptor activation.

## Results

### Production of biologically active Norrin

To address the challenge of producing Norrin in large quantities, we screened conditions and constructs for Norrin expression (*Figure 1A*). We found that fusion of Norrin to the C-terminus of small ubiquitin-like

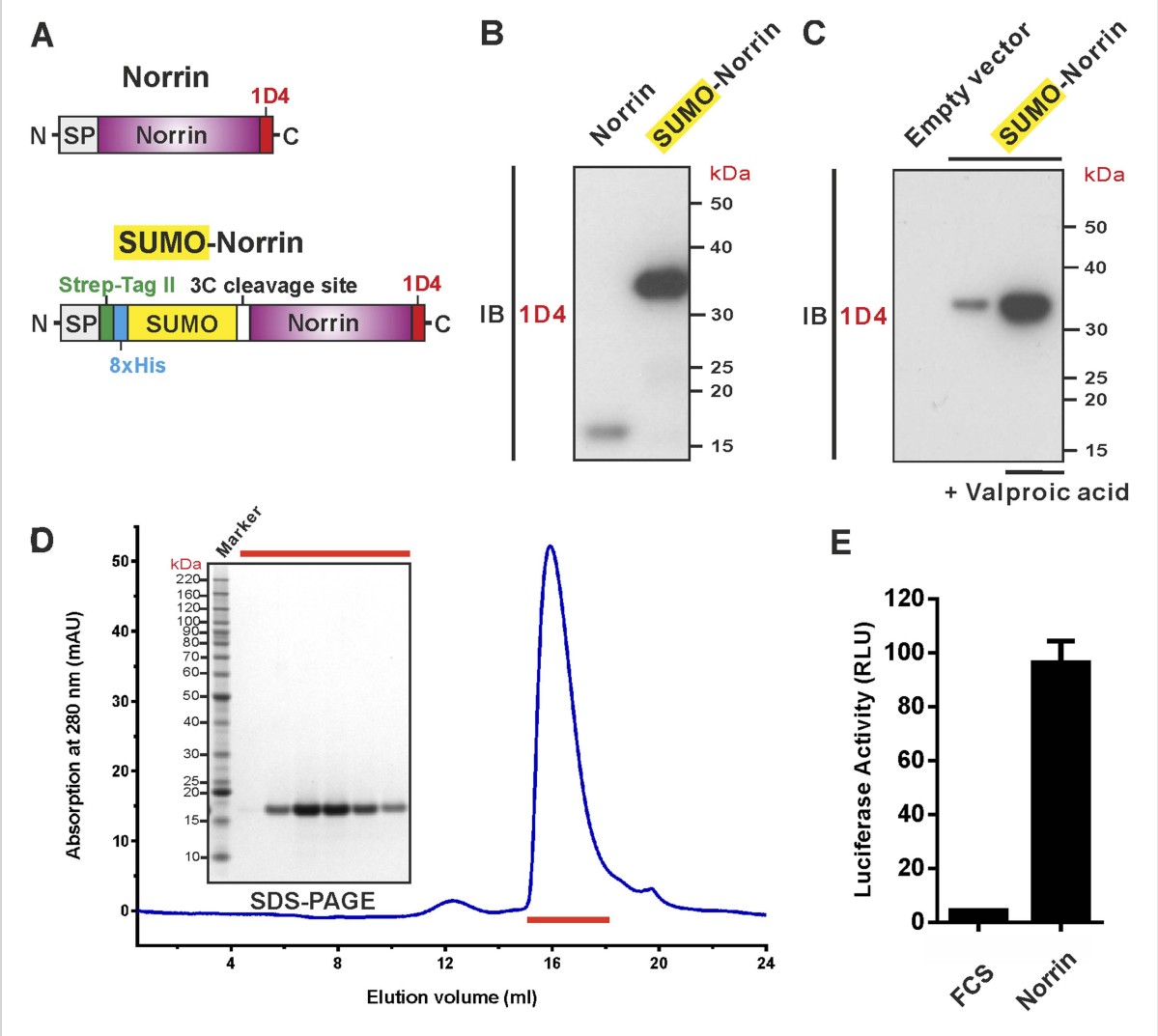

**Figure 1**. Expression and purification of biologically active recombinant Norrin. (**A**) Schematic diagrams of the expression constructs including Norrin (a signal peptide, SP, followed by Norrin and Rho-1D4 tag at C-terminus) and SUMO-Norrin (a SP followed by a Strep-tag II, an octahistidine, SUMO, HRV 3C protease cleavage site, Norrin, and Rho-1D4 tag at C-terminus). (**B** and **C**) Conditioned media from transfected HEK293T cells were immunoblotted (IB) with the anti-Rho-1D4 antibody. (**B**) SUMO fusion improves Norrin secreted expression. (**C**) The expression level of SUMO tagged Norrin was further boosted for HEK-293T cells treated with valproic acid. (**D**) SEC elution profile and SDS-PAGE under reducing conditions with fractions analysed marked by red lines. (**E**) Purified recombinant untagged Norrin activates the canonical Wnt/β-catenin pathway in the luciferase reporter assay. RLU: relative light unit. Error bars indicate standard deviations (n = 3).

modifier (SUMO) (*Peroutka et al., 2008*), in combination with addition of valproic acid (*Backliwal et al., 2008*), a putative histone deacetylase inhibitor, substantially boosted expression of the secreted protein in human embryonic kidney (HEK) 293T cells (*Figure 1B,C*). After removal of the SUMO fusion tag, the recombinant Norrin shows a monodispersed state in size-exclusion chromatography (SEC; *Figure 1D*) and is biologically active in a cell-based luciferase reporter assay (*Figure 1E*).

## The crystal structure of Norrin and its oligomeric state in solution

We determined three crystal structures of Norrin (*Figure 2A* and *Table 1*), using selenomethionine-labeled protein for phasing (*Figure 2—figure supplement 1*). The Norrin protein fold is identical to that of the previously reported MBP-Norrin crystal structure (*Ke et al., 2013*). Each Norrin monomer comprises three β-hairpins (β1-β2, β3-β4 and β5-β6), a β7 strand at the C-terminus, and four

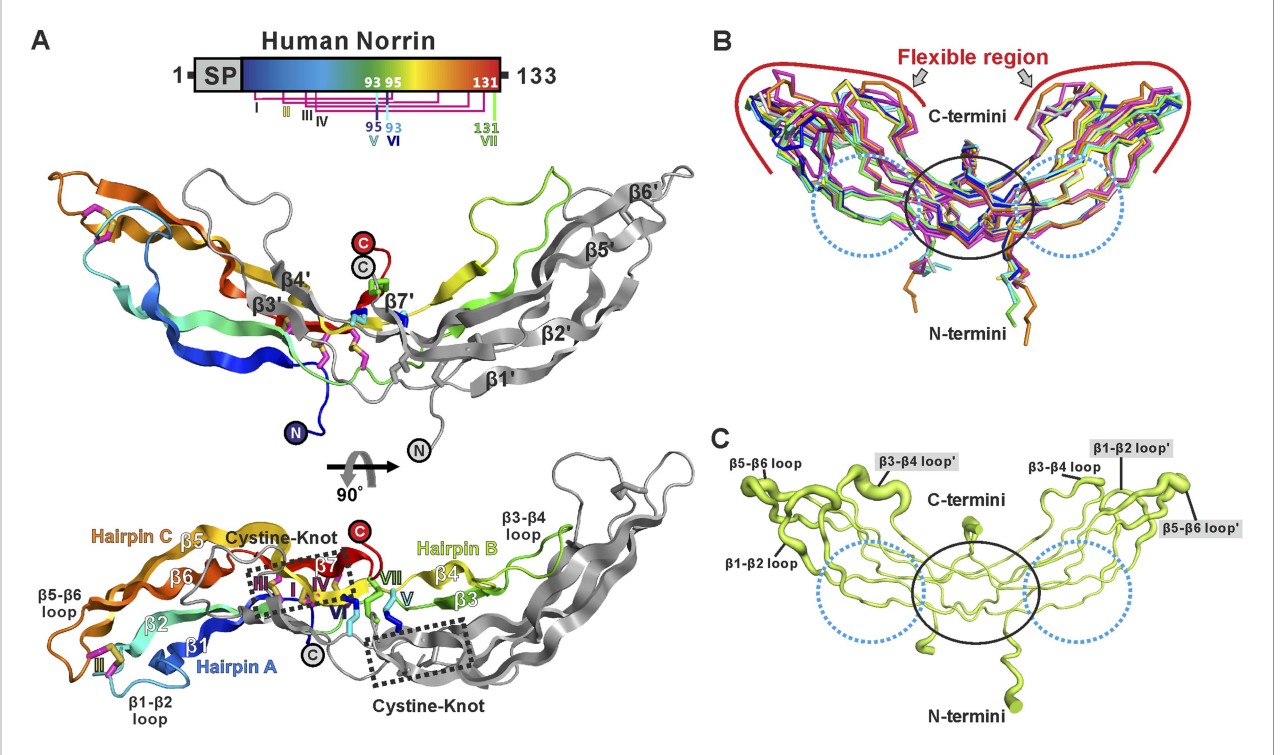

**Figure 2**. Crystal structure and structural analysis of apo Norrin. (**A**) Schematic diagram of Norrin is rainbow coloured and disulphide bonds are drawn as lines. Cartoon representation of dimeric Norrin. Four intramolecular disulphide bonds are shown as magenta sticks. Cys93, Cys95, and Cys131 (forming intermolecular disulphide bridges) are shown as cyan, blue, and green sticks, respectively. Two cystine-knot motifs are marked with dotted boxes and the filled circles denote the N- and C-termini. (**B**) Ribbon diagram of superpositions of Norrin molecules from the asymmetric unit of crystal form I (green, chain A and B; cyan, chain C and D; magenta, chain E and F), crystal form II (yellow, chain A and B; blue, chain C and D), crystal form III (grey, chain A and B; purple, chain C and D), and MBP-Norrin (cyan; PDB ID: 4MY2). The flexible regions are highlighted as red lines. Loop regions (β1-β2 loop, β3-β4 loop, and β5-β6 loop) show structural plasticity. The well ordered regions include the cystine-knot motifs plus intermolecular disulphide linked areas (black circle) and two Fz4 binding sites (cyan dotted circle). (**C**) Representative Norrin dimer displayed with the diameter of the Cα tube defined by Cα atom B factor (small tube means structural rigidity; large tube indicates structural flexibility).

The following figure supplements are available for figure 2:

**Figure supplement 1**. Electron density map of Norrin structure.

**Figure supplement 2**. Multiple sequence alignment of Norrin.

**Figure supplement 3**. Norrin solution structure and structural analyses.

intramolecular disulphide bonds (*Figure 2—figure supplement 2*). The two monomers assemble as an elongated, head-to-tail, dimer (*Figure 2A*) stabilized by three intermolecular disulphide bridges (Cys93-Cys95, Cys95-Cys93, and Cys131-Cys131), in agreement with small-angle X-ray scattering (SAXS) measurements which showed Norrin dimer in solution (*Figure 2—figure supplement 3A* and *Table 2*). The dimer interface is further stabilized by extensive hydrogen bonds and hydrophobic interactions (*Figure 2—figure supplement 3B,C*). Superposition of all molecules in the asymmetric units from our three crystal forms with the MBP-Norrin structure (*Ke et al., 2013*) showed an average root-mean-square (r.m.s.) deviation of 1.5 Å over 190 equivalent Cα atoms (*Figure 2—figure supplement 3D*). On inspection the superpositions revealed a high degree of conformational plasticity in the β1-β2, β3-β4 and β5-β6 loops (*Figure 2B*). The flexibility inherent in these regions is consistent with the relatively high crystallographic B factor values (*Figure 2C*). Conversely, the structural comparisons underscore the conserved nature of the interface at the dimer core. It has previously been noted that disruption of the dimer by either Cysteine-to-Alanine mutations of

**Table 1.** Data collection, phasing and refinement statistics

| | Norrin–Fz4$_{CRD}$–SOS | Methylated Norrin–Fz4$_{CRD}$ | Norrin | Norrin Se-Met | Methylated Norrin |
|---|---|---|---|---|---|
| **Crystal form** | | | I | I | II |
| Data collection | | | | | |
| Space group | $P6_122$ | $P4_322$ | $P2_12_12_1$ | $P2_12_12_1$ | $P2_12_12_1$ |
| Cell dimensions | | | | | |
| $a, b, c$ (Å) | 119.1, 119.1, 119.2 | 98.9, 98.9, 120.4 | 46.4, 79.1, 243.3 | 45.8, 78.8, 232.8 | 102.7, 53.1, 96.1 |
| $\alpha, \beta, \gamma$ (°) | 90, 90, 90 | 90, 90, 120 | 90, 90, 90 | 90, 90, 90 | 90, 90, 90 |
| | | | | Peak | |
| Wavelength | 0.9200 | 0.9795 | 0.9686 | 0.9795 | 0.9795 |
| Resolution (Å) | 47.34–3.00 (3.18–3.00) | 49.46–2.30 (2.38–2.30) | 65.56–2.40 (2.49–2.40) | 116.39–3.18 (3.26–3.18) | 33.65–2.00 (2.05–2.00) |
| $R_{pim}$ (%) | 3.1 (54.8) | 4.5 (56.1) | 6.1 (42.3) | 2.8 (23.4) | 4.1 (58.3) |
| $I/\sigma I$ | 14.6 (1.6) | 10.7 (1.4) | 7.8 (1.9) | 20.2 (3.0) | 9.1 (1.7) |
| Completeness (%) | 100 (100) | 98.9 (97.2) | 99.9 (100) | 99.9 (99.9) | 100 (100) |
| Redundancy | 19.6 (20.6) | 6.0 (5.6) | 5.6 (5.7) | 33.3 (9.9) | 5.6 (5.8) |
| Refinement | | | | | |
| Resolution (Å) | 47.34–3.00 (3.18–3.00) | 49.46–2.30 (2.38–2.30) | 65.56–2.40 (2.49–2.40) | | 33.65–2.00 (2.05–2.00) |
| No. reflections | 10,503 (1648) | 26,816 (2514) | 34,722 (3384) | | 36,272 (2635) |
| $R_{work}/R_{free}$ | 21.5/26.7 | 19.7/22.1 | 21.6/26.2 | | 23.3/24.8 |
| No. atoms | | | | | |
| Protein | 1759 | 2557 | 4930 | | 3187 |
| Ligand/ion | 83 | 39 | 101 | | 10 |
| Water | 0 | 115 | 164 | | 122 |
| $B$-factors | | | | | |
| Protein | 113 | 63 | 70 | | 57 |
| Ligand/ion | 133 | 71 | 92 | | 73 |
| Water | 0 | 57 | 55 | | 51 |
| R.m.s deviations | | | | | |
| Bond lengths (Å) | 0.005 | 0.004 | 0.009 | | 0.005 |
| Bond angles (°) | 1.18 | 0.93 | 1.08 | | 1.07 |
| Ramachandran plot | | | | | |
| Favored (%) | 95.5 | 97.0 | 96.7 | | 97.2 |
| Allowed (%) | 4.5 | 3.0 | 3.3 | | 2.8 |
| PDB code | 5BQC | 5BQE | 5BPU | | 5BQ8 |

| | Norrin | Fz4$_{CRD}$ | Fz4$_{CRD}$ |
|---|---|---|---|
| **Crystal form** | III | I | II |
| Data collection | | | |
| Space group | $C121$ | $P2_12_12_1$ | $P6_1$ |
| Cell dimensions | | | |
| $a, b, c$ (Å) | 86.8, 38.1, 177.2 | 72.6, 102.1, 116.5 | 76.1, 76.1, 204.5 |
| $\alpha, \beta, \gamma$ (°) | 90, 94, 90 | 90, 90, 90 | 90, 90, 90 |
| Wavelength | 0.9795 | 0.9686 | 0.9686 |
| Resolution (Å) | 44.19–2.30 (2.38–2.30) | 41.77–2.20 (2.27–2.20) | 47.37–2.40 (2.49–2.40) |

*Table 1. Continued on next page*

*Table 1. Continued*

|  | Norrin | Fz4$_{CRD}$ | Fz4$_{CRD}$ |
|---|---|---|---|
| **Crystal form** | III | I | II |
| $R_{pim}$ (%) | 2.8 (36) | 4.1 (49.5) | 2.6 (33.9) |
| $I/\sigma I$ | 16.7 (2.0) | 12.8 (2.0) | 14.5 (2.2) |
| Completeness (%) | 99.2 (97.7) | 99.2 (99.7) | 99.5 (99.4) |
| Redundancy | 5.8 (6.0) | 4.3 (4.4) | 4.0 (4.1) |
| Refinement |  |  |  |
| Resolution (Å) | 44.19–2.30 (2.38–2.30) | 41.77–2.20 (2.27–2.20) | 47.37–2.40 (2.49–2.40) |
| No. reflections | 26,073 (2538) | 44,268 (3802) | 25,975 (2724) |
| $R_{work}/R_{free}$ | 22.1/25.0 | 17.7/22.3 | 20.3/24.3 |
| No. atoms |  |  |  |
| Protein | 3104 | 3866 | 3877 |
| Ligand/ion | 72 | 70 | 99 |
| Water | 54 | 148 | 69 |
| $B$-factors |  |  |  |
| Protein | 91 | 47 | 76 |
| Ligand/ion | 72 | 67 | 72 |
| Water | 142 | 43 | 68 |
| R.m.s deviations |  |  |  |
| Bond lengths (Å) | 0.006 | 0.01 | 0.005 |
| Bond angles (°) | 1.03 | 1.35 | 0.94 |
| Ramachandran plot |  |  |  |
| Favored (%) | 96.0 | 99.0 | 97.0 |
| Allowed (%) | 4.0 | 1.0 | 3.0 |
| PDB code | 5BQB | 5BPB | 5BPQ |

All structures were determined from one crystal.

Values in parentheses are for highest-resolution shell.

intermolecular disulphide bonds or mutations of hydrophobic residues at the dimer interface results in a loss of Norrin-mediated signalling (*Smallwood et al., 2007*; *Ke et al., 2013*).

## The crystal structure of Fz4$_{CRD}$

We determined two crystal structures of Fz4$_{CRD}$ (*Figure 3* and *Table 1*). Similar to mouse Fz8$_{CRD}$ (*Dann et al., 2001*) the Fz4$_{CRD}$ fold comprises four α helices (*Figure 3A* and *Figure 3—figure supplement 1A*) stabilized by five disulphide bridges (Cys45–Cys106, Cys53–Cys99, Cys90–Cys128, Cys117–Cys158, Cys121–Cys145). The N-acetylglucosamines on two N-linked glycosylation sites at Asn59 and Asn144 are visible in the electron density map (*Figure 3A*). Superposition of all Fz4$_{CRD}$ molecules in the asymmetric units from two crystal forms revealed a well-ordered protein fold (*Figure 3—figure supplement 1B*). The conserved disulphide bonds in Fz$_{CRD}$ superfamily members are essential for functional activity. Familial Exudative Vitreoretinopathy disease mutant C45Y results in misfolded protein retained in the endoplasmic reticulum, similar to the effects of Cysteine-to-Alanine mutations in the related CRD of *Drosophila* Smoothened (Smo) (*Zhang et al., 2011*; *Rana et al., 2013*). Structural comparison showed Fz4$_{CRD}$ closely resembles the CRDs of mouse Fz8 and secreted Frizzled-related protein 3 (sFRP3) with an average r.m.s. deviation of 1.2 Å over 115 equivalent Cα atoms (*Figure 3—figure supplement 1C–E*) and approximate sequence identity of 35%. Comparisons with the CRDs of muscle-specific kinase (MuSk) and Smo showed more substantial structural differences with an average r.m.s. deviation of 2.3 Å over 86 equivalent Cα atoms (*Figure 3—figure supplement 1F–H*).

**Table 2**. Molecular properties of the proteins determined by SAXS

| Proteins | N-Glyc state | $R_g$ (nm)* | $D_{max}$ (nm)† | Volume porod ($V_p$ [nm³]) | MW$_{Theoretical}$ (kDa)‡ | MW$_{Measured}$ (KDa)§ | MW$_{Measured}$ (KDa)# |
|---|---|---|---|---|---|---|---|
| Fz4$_{CRD}$ | deglyc¶ | 1.98 | 6.93 | 33.0 | 17.1 (monomer) | 15.9 | 19.9 |
| Fz4$_{CRD}$ | glyc** | 2.24 | 7.84 | 41.1 | 21.4 (monomer) | 23.7 | 24.7 |
| Norrin | | 2.74 | 9.18 | 37.4 | 27.2 (dimer) | 33.5 | 22.5 |
| Norrin–Fz4$_{CRD}$ | deglyc¶ | 3.41 | 11.92 | 93.8 | 61.3 (2:2 complex) | 57.9 | 56.5 |

*$R_g$ is Radius of gyration, calculated from Guinier plot using AutoRg (**Petoukhov et al., 2012**).

†$D_{max}$ is the maximum dimension of the particle, calculated by GNOM (**Svergun, 1992**).

‡The theoretical molecular weight (MW$_{Theoretical}$) is predicated from amino acid sequence plus the molecular weight of N-linked glycans (see 'Materials and methods', **SEC-MELS analysis** for detailed information of calculation).

§The measured molecular weight (MW$_{Measured}$) is calculated from forward scattering of sample ($I(0)$) by comparison with reference bovine serum albumin (BSA).

#The measured molecular weight (MW$_{Measured}$) is obtained by dividing the Volume Porod ($V_p$ [nm³]) by 1.66 (**Rambo and Tainer, 2011**).

¶The proteins were produced from HEK293T cells in the presence of kifunensine with limited glycosylation and treated with endoglycosidase-F$_1$.

**The proteins were produced from HEK293T cells with full glycosylation.

## Assessment of the monomeric states of Fz$_{CRD}$ in solution

Fz receptors are members of the GPCR family (**Nichols et al., 2013**), known for formation of receptor dimers, although it is unclear whether dimerization is mediated by the CRD, transmembrane helices or intracellular domain. In the case of Fz4, β-galactosidase complementation in combination with bioluminescence resonance energy transfer and split-yellow fluorescence protein assays suggest that Fz4 exists as dimer on the cell membrane in the absence of Norrin or Wnts (**Kaykas et al., 2004**; **Ke et al., 2013**). However, ligand-independent receptor dimerization of Fz4 is not sufficient to activate signalling (**Xu et al., 2004**; **Ke et al., 2013**). Interestingly, we found that our Fz4$_{CRD}$ structures form the same dimeric assembly in two crystal lattices (r.m.s. deviation of 0.7 Å over 238 equivalent Cα atoms from two crystal forms; **Figure 3—figure supplement 2A**). The dimer interface has an average 1330 Å² buried surface area, in agreement with the characteristics of known protein–protein interfaces (**Lawrence and Colman, 1993**). However, this Fz4$_{CRD}$ dimer (front-to-front) is distinct from the previously reported crystal structure of mouse Fz8$_{CRD}$ dimer (back-to-back; **Figure 3—figure supplement 2B**) (**Dann et al., 2001**). We were therefore curious to assess the dimerization characteristics of the CRDs of Fz receptors. Size-exclusion chromatography coupled to multi-angle light scattering (SEC-MALS) results (**Figure 3B** and **Table 3**) showed Fz4$_{CRD}$, Fz5$_{CRD}$ and Fz8$_{CRD}$ exist as monomers in solution at 50 μM concentration, in agreement with previously reported SEC studies of Fz8$_{CRD}$ and SEC-MALS analyses of MuSK$_{CRD}$ and Smo$_{CRD}$ (**Stiegler et al., 2009**; **Nachtergaele et al., 2013**). SAXS measurements further support the conclusion that Fz4$_{CRD}$ is monomeric in solution at 290 μM concentration (**Figure 3C,D**). Taken together, our results suggest that the CRDs of Fz receptors exist as monomers and may not be involved in receptor dimerization; multiple GPCRs dimerize through their hepta-helical transmembrane domains (**Rios et al., 2001**). However, we cannot exclude the possibility that in the environment of the cellular membrane the weak interaction propensities of the CRDs, in combination with the transmembrane domains, are important for the dimerization of Fz receptors.

## The crystal structure of Norrin in complex with Fz4$_{CRD}$

We purified Norrin–Fz4$_{CRD}$ complex (**Figure 4—figure supplement 1A**) and determined the crystal structures of methylated Norrin–Fz4$_{CRD}$ (dimethylated surface-exposed lysine residues; **Figure 4—figure supplement 1B,C**) and Norrin–Fz4$_{CRD}$–SOS (complex bound with heparin mimic sucrose octasulfate, SOS; **Figure 4A** and **Figure 4—figure supplement 1D**) at 2.3 Å and 3.0 Å resolution, respectively (**Table 1**). These two complex structures show different stoichiometries: a 2:1 complex for the methylated Norrin–Fz4$_{CRD}$ and a 2:2:2 stoichiometry for the Norrin–Fz4$_{CRD}$–SOS complex, the architecture of which resembles a butterfly (**Figure 4A**). The Norrin–Fz4$_{CRD}$ binding interface is conserved between the complex structures (**Figure 4—figure supplement 1E**). Each Fz4$_{CRD}$ interacts one-to-one with a separate Norrin chain, burying on average 1680 Å² of surface area.

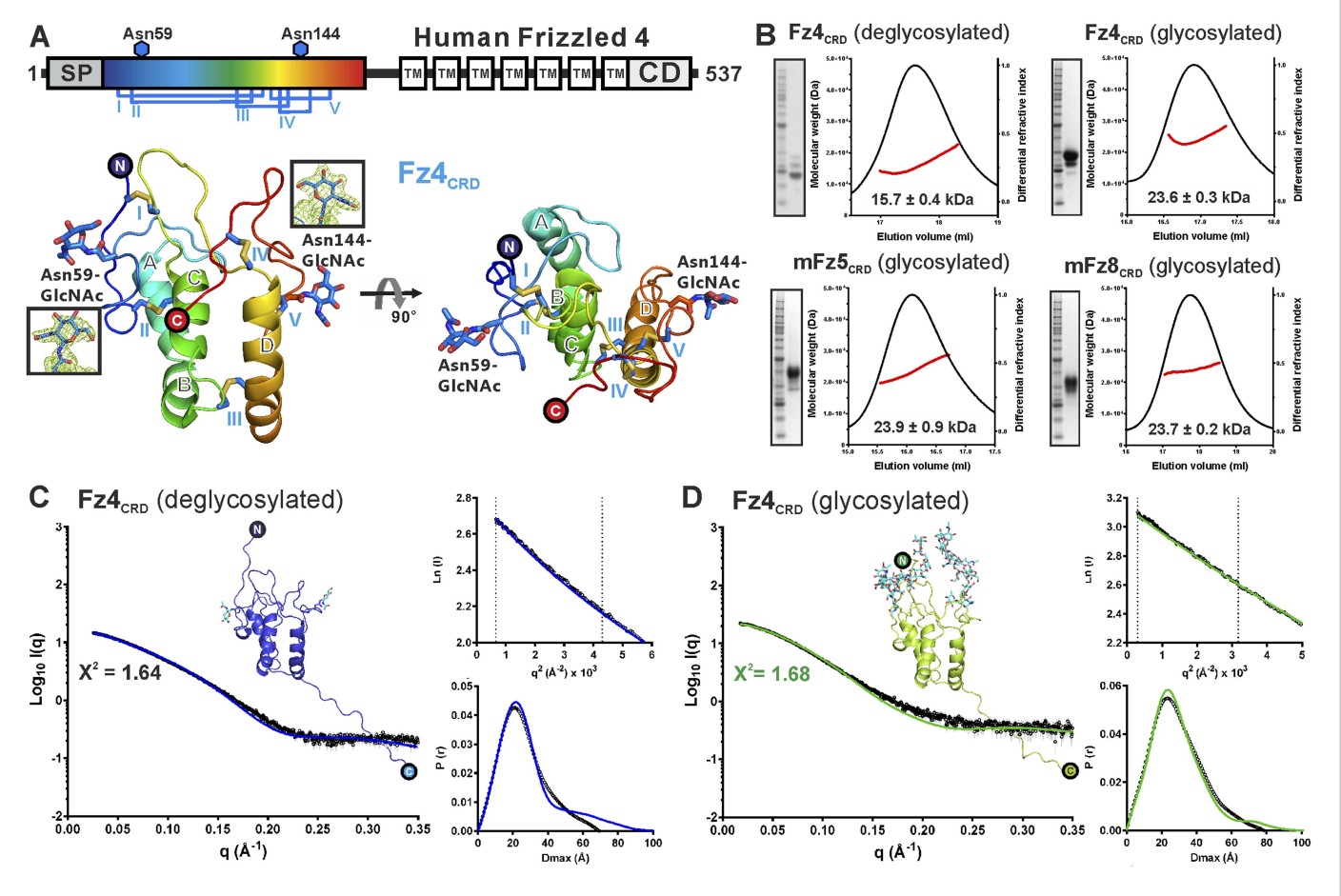

**Figure 3**. Crystal and solution structures of unliganded Fz4$_{CRD}$. (**A**) Schematic domain organization (SP, signal peptide; TM. transmembrane domain; CD, cytoplasmic domain). Crystallization constructs are rainbow coloured. Disulphide bonds are drawn and blue hexagons denote N-linked glycosylation sites. Cartoon representation of Fz4$_{CRD}$ in rainbow colouring. N-linked N-acetyl-glucosamines (GlcNAc) and disulphide bonds are shown as blue sticks. (**B**) SEC-MALS experiments. The red line represents the molecular weight (left ordinate axis) and black lines show the differential refractive index (right ordinate axis) as well as SDS-PAGE (Inset). The numbers denote the corresponding molecular weights of each peak. (**C** and **D**) SAXS analyses of deglycosylated and glycosylated Fz4$_{CRD}$ solution structures. The experimental scattering data (black circles) and calculated scattering patterns (coloured lines) are shown and the Fz4$_{CRD}$ solution structure model is presented. The upper right inset shows the experimental (black circles) and calculated (coloured lines) Guinier region. The dashed lines delimit the range of fitting for $R_g$ analysis ($R_g \cdot S \leq 1.3$). The bottom right inset shows the experimental (black circles) and calculated (coloured lines) pair distance distribution $P(r)$ curve.

The following figure supplements are available for figure 3:

**Figure supplement 1**. Multiple sequence alignment and structural analysis of cysteine-rich like domains.

**Figure supplement 2**. Distinct dimeric assembly of Fz4$_{CRD}$ and mouse Fz8$_{CRD}$ observed from crystal structures.

To investigate the preference for complex formation in a 2:1 or 2:2 stoichiometry, we performed SEC-MALS (*Figure 4B*) and SAXS (*Figure 4C*) measurements in the absence of SOS. Both methods show Norrin interacts with Fz4$_{CRD}$ in a 2:2 stoichiometry. Lysine methylation of the Norrin–Fz4$_{CRD}$ complex was used to facilitate crystal lattice formation (*Walter et al., 2006*; *Malinauskas et al., 2011*), and on close inspection of the structure we found Lys102 and Lys104, two residues which contribute to the Norrin–Fz4$_{CRD}$ interface, (see next section) are dimethylated in the uncomplexed subunit of the Norrin dimer (*Figure 4—figure supplement 1C*), and contribute instead to a lattice contact. This observation suggests that the 2:1 stoichiometry merely reflects the favourable crystallization characteristics of a sub population of asymmetrically methylated Norrin–Fz4$_{CRD}$ complexes. Thus although the

Table 3. Molecular properties of the proteins determined by SEC-MALS

| Protein | Number of N-glyc sites | N-Glyc state | MW$_{Theoretical}$ (kDa)‡ | MW$_{Measured}$ (KDa) |
|---|---|---|---|---|
| Fz4$_{CRD}$ | 2 | deglyc* | 17.1 (monomer) | 15.7 ± 0.4 |
| Fz4$_{CRD}$ | 2 | glyc† | 21.4 (monomer) | 23.6 ± 0.3 |
| mFz5$_{CRD}$ | 2 | glyc† | 22.2 (monomer) | 23.9 ± 0.9 |
| mFz8$_{CRD}$ | 2 | glyc† | 22.1 (monomer) | 23.7 ± 0.2 |
| Norrin–Fz4$_{CRD}$ | 4 (2:2 complex) | deglyc* | 61.3 (2:2 complex) | 60.1 ± 0.4 |
| Norrin–Fz4$_{CRD}$ | 4 (2:2 complex) | glyc† | 69.9 (2:2 complex) | 61.3 ± 0.5 |

*The proteins were produced from HEK293T cells in the presence of the N-glycosylation processing inhibitors, kifunensine resulting in limited glycosylation and were treated with endoglycosidase-F$_1$.
†The proteins were produced from HEK293T cells with full glycosylation.
‡The measured molecular weight (MW$_{Measured}$) is in general agreement with theoretical molecular weight (MW$_{Theoretical}$) predicated based on the primary sequence plus the molecular weight of N-linked glycans (see 'Materials and methods', **SEC-MELS analysis** for detailed information of calculation).

methylated Norrin–Fz4$_{CRD}$ structure usefully provides high-resolution information for the ligand–receptor interface (*Figure 4—figure supplement 1F*), the Norrin–Fz4$_{CRD}$–SOS structure defines the overall architecture of the native complex (*Figure 4A*). The two Fz4$_{CRD}$ diverge from the Norrin dimer without contacting each other (*Figure 4A*), and with their C-termini suitably oriented for attachment to the same cell surface.

The Norrin–Fz4$_{CRD}$ complex has a novel architecture; the mode of interaction of Norrin is distinct from that of other cystine-knot secreted growth factors (transforming growth factor-β, bone morphogenetic protein, platelet-derived growth factor, and vascular endothelial growth factor) with either their receptors or antagonists (*Figure 4—figure supplement 2*).

Neither Norrin nor Fz4$_{CRD}$ undergoes large conformational changes upon complex formation, although the flexibility of residues involved in the binding interface is reduced (*Figure 4—figure supplement 3*). Interestingly, superposition of the Norrin–Fz4$_{CRD}$ complex and the previously reported MBP-Norrin structure resulted in steric clashes between the Fz4$_{CRD}$ and the MBP (*Figure 4—figure supplement 4*). This suggests that MBP hinders Norrin interaction with Fz4$_{CRD}$ consistent with MBP-Norrin only having half of the signalling activity of untagged Norrin (*Ke et al., 2013*).

## Analyses of binding interfaces

At the ligand–receptor interface (*Figure 5A*) two β-hairpins in Norrin (β1-β2 and β5-β6) contact three loops in Fz4$_{CRD}$ (I, II, and III). Fz4$_{CRD}$ loop I hydrogen bonds to Norrin (*Figure 5B*). Fz4$_{CRD}$ loop II makes extensive hydrophobic contacts plus one salt-bridge (Fz4$_{CRD}$ Lys109 with Norrin Asp46; *Figure 5C*). Fz4$_{CRD}$ loop III interacts with Norrin via an extensive hydrogen bond network as well as hydrophobic contacts (*Figure 5D*). Interactions with SOS involve the positively charged residues of Lys58, Arg107, Arg109, and Arg115 on Norrin, plus His154 and Asn155 on Fz4$_{CRD}$ loop III (*Figure 5E*). These residues define a likely binding site for GAGs, in agreement with previous reports of Norrin interactions with extracellular matrix and heparin (*Xu et al., 2004*; *Ohlmann et al., 2010*).

## Verification of Fz4 binding site

The Norrin–Fz4 interface revealed in our crystal structures (*Figure 6A,B*) is in excellent agreement with reported disease-associated mutations (*Figure 6C*) and surface residue conservation (*Figure 6D*). We performed mutagenesis and functional assays to verify this Fz4 binding site. Surface plasmon resonance (SPR) experiments (*Figure 6E* and *Figure 6—figure supplement 1A*) show a micromolar equilibrium dissociation constant between Norrin and Fz4$_{CRD}$. Mutations of either H43N/V45T or L61N/A63S, which resulted in the introduction of an N-linked glycosylation site in the Fz4 binding site on Norrin, completely abolish the interaction (*Figure 6—figure supplement 1B*). Norrin disease-associated mutants V45E and L61P/A63D lose binding affinity for Fz4$_{CRD}$ (*Figure 6—figure supplement 1C*), as do mutants R41E/H43E and R38E/R41S/H43E/K102E/K104E (*Figure 6—figure supplement 1D*). In contrast, Norrin mutants L52N/K54S and M114N/L116S (to introduce an N-linked

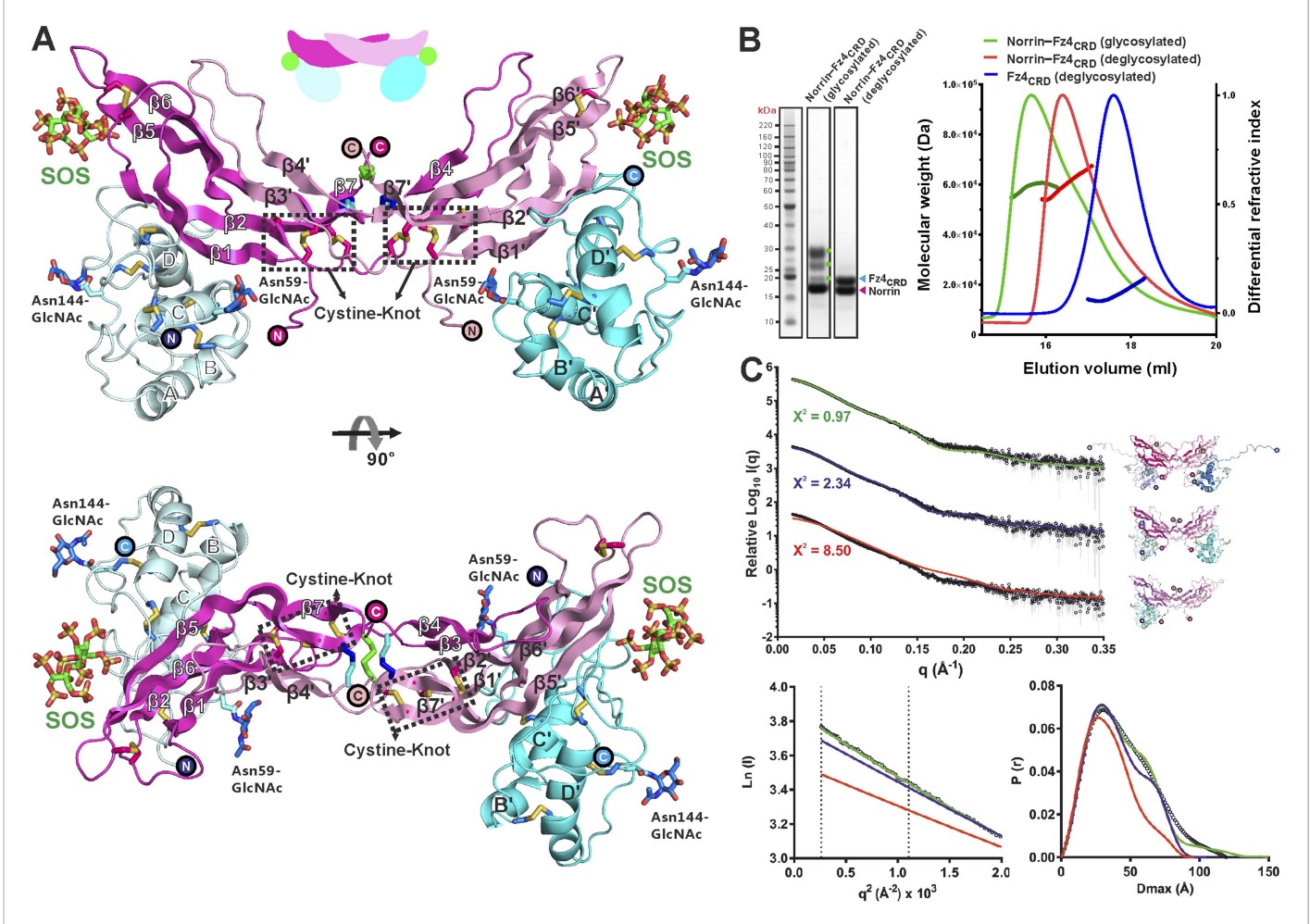

**Figure 4**. Crystal structure and solution behaviour of Norrin–Fz4$_{CRD}$ complex. (**A**) Ribbon representation of Norrin (magenta and pink) in a 2:2:2 complex with Fz4$_{CRD}$ (cyan and pale cyan) and SOS (green). (**B**) SEC-MALS analyses. The profile of molecular weight (left ordinate axis) and differential refractive index (right ordinate axis) are shown as thick and thin lines, respectively. SDS-PAGE (Inset) shows Norrin in complex with Fz4$_{CRD}$ (triplet band for glycosylated Fz4$_{CRD}$, marked as green circles, represents glycosylation heterogeneity). (**C**) SAXS experiments. Experimental scattering data (black circles) and calculated scattering patterns (coloured lines) are shown to a maximal momentum transfer of q = 0.35 Å$^{-1}$. Individual data: fit pairs are displaced along an arbitrary y axis to allow for better visualization. Bottom curve: Norrin–Fz4$_{CRD}$ 1:2 complex crystal structure (red line). Middle curve: Norrin–Fz4$_{CRD}$ 2:2 complex crystal structure (blue line). Top curve: modelled Norrin–Fz4$_{CRD}$ 2:2 complex crystal structure (missing regions for Norrin and Fz4$_{CRD}$ N- and C-termini are modeled into the crystal complex structure; green line). Structural models are shown in cartoon representation. The bottom left inset shows the experimental (black circles) and calculated (coloured lines) Guinier region. The bottom right inset shows the experimental (black circles) and calculated (coloured lines) pair distance distribution $P$(r) curves.

The following figure supplements are available for figure 4:

**Figure supplement 1**. Protein complex production and structural properties of Norrin–Fz4$_{CRD}$ complex.

**Figure supplement 2**. Structural comparison of cystine-knot growth factor monomers and their ternary complexes.

**Figure supplement 3**. No large conformational changes upon complex formation.

**Figure supplement 4**. Structural comparison of Norrin–Fz4$_{CRD}$ complex with MBP-Norrin.

glycosylation site in the β1-β2 loop or β5-β6 loop, respectively), predicted to lie outside the Fz4 binding site (*Figure 6A*), show the same binding affinity as wild-type (*Figure 6—figure supplement 1E*). Cell-based Wnt/β-catenin responsive luciferase assays (*Figure 6F*) further support the significance of

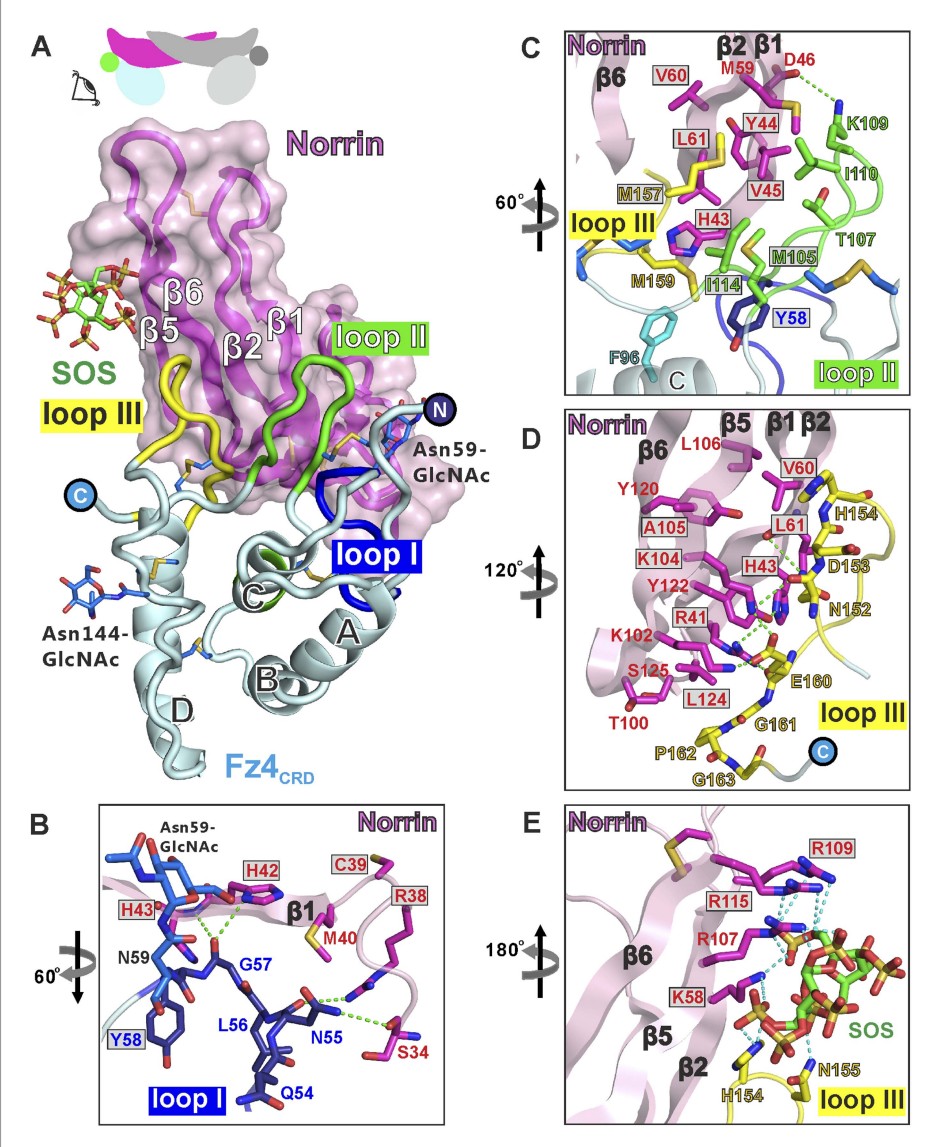

**Figure 5**. Structural details of binding sites in the Norrin–Fz4$_{CRD}$–SOS complex. (**A**) Side-view of complex. Fz4$_{CRD}$ loops involved in Norrin binding are coloured blue (loop I), green (loop II), and yellow (loop III). (**B–E**) Views detailing the interfaces. Selected residues involved in binding are shown as sticks and coloured magenta (Norrin), blue (loop I), green (loop II), yellow (loop III), and cyan (Phe96 of Fz4$_{CRD}$) and those associated with disease mutations are highlighted in boxes. Dotted lines denote hydrogen bonds. (**B**) Interactions between Fz4$_{CRD}$ loop I and Norrin. (**C**) Hydrophobic interactions of Norrin with Fz4$_{CRD}$ loop II and part of loop III. (**D**) Interactions of Fz4$_{CRD}$ loop III with Norrin. (**E**) SOS binding to Norrin and Fz4$_{CRD}$ loop III.

the Fz4 binding site. Norrin mutants that lose binding to Fz4$_{CRD}$ also fail to induce the luciferase reporter activity, in agreement with the SPR results (*Figure 6—figure supplement 1*) and prior genetic data (*Xu et al., 2004*; *Smallwood et al., 2007*). Taken together, our structural and functional results suggest that Norrin uses β strands (β1-β2 and β5-β6) for Fz4$_{CRD}$ binding rather than, as proposed by *Bazan et al. (2012)* using the loop between β1 and β2 (*Bazan et al., 2012*).

To determine the binding affinity of Norrin for different CRD of Fz receptors, we undertook a series of SPR experiments. The results (*Figure 6—figure supplement 1F*) show that Norrin has greatest affinity for Fz4$_{CRD}$ (Kd: 1 μM), low affinities for Fz5$_{CRD}$ (Kd: 42 μM) and Fz8$_{CRD}$ (Kd: 64 μM), and no binding to Fz7$_{CRD}$. In combination, these results confirm that pairing Norrin with Fz4$_{CRD}$ provides

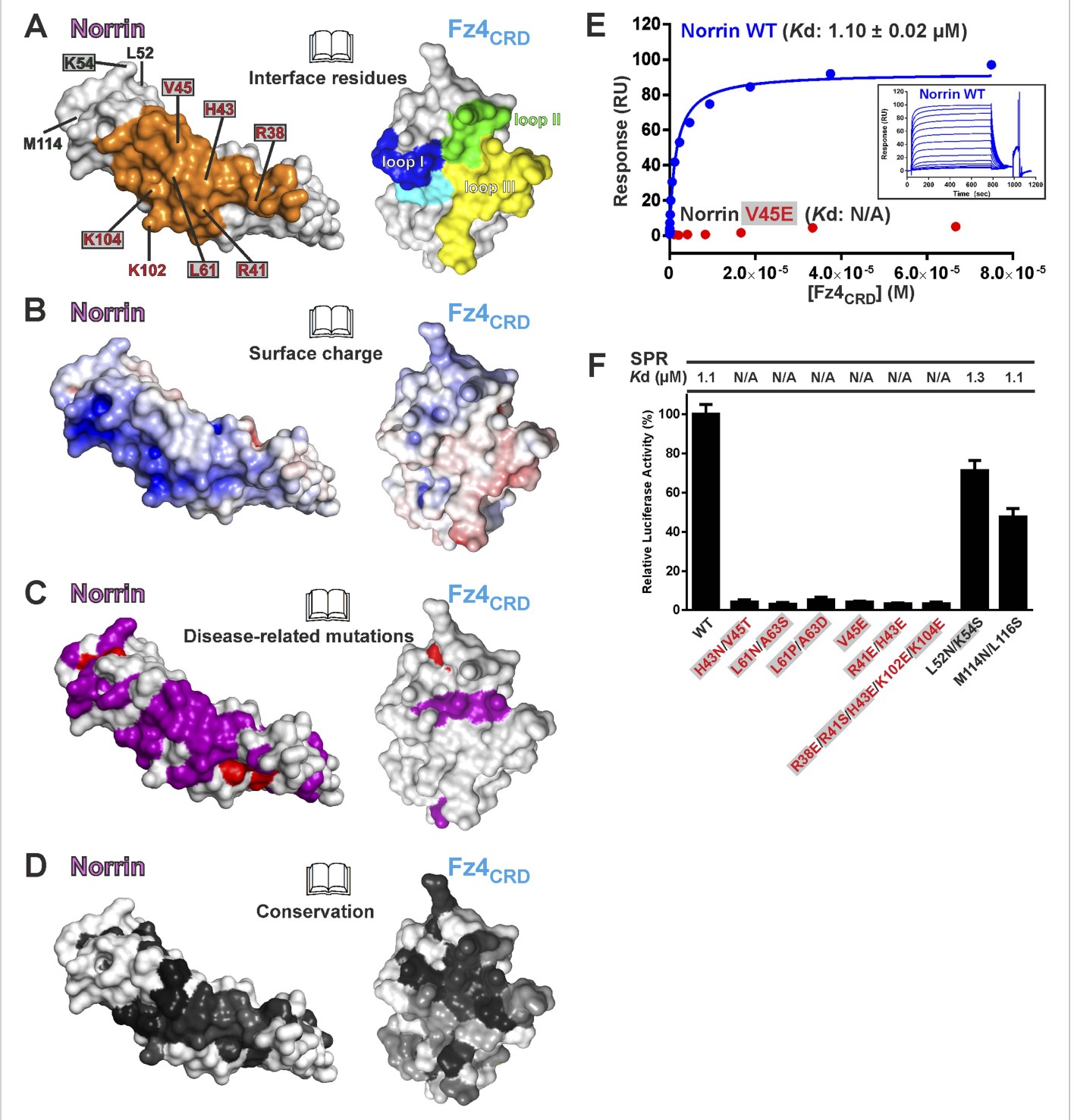

**Figure 6**. Biophysical and functional characterisation of Fz4 binding site. Surface representation of Norrin–Fz4$_{CRD}$ complex in open book view. (**A**) Interface residues are coloured orange (Norrin) and blue (loop I), green (loop II), yellow (loop III), and cyan (Phe96) on Fz4$_{CRD}$. Norrin mutation sites used in functional assays are labelled (red, residues involved in Fz4$_{CRD}$ binding; grey filled box, residues associated with diseases; black, residues located outside the Fz4 binding site). (**B**) Norrin and Fz4$_{CRD}$ coloured by electrostatic potential from red (acidic; $-7\ k_bT/e_c$) to blue (basic; $7\ k_bT/e_c$). (**C**) Disease-associated mutations mapped onto the surface of Norrin and Fz4$_{CRD}$ (purple, missense mutations; red, missense mutations of cysteine residues). (**D**) Surfaces colour-coded according to sequence conservation from white (not conserved) to black (conserved). (**E**) SPR results for Fz4$_{CRD}$ binding to Norrin wild-type (WT) and Norrin V45E mutant. Inset SPR sensorgrams are of equilibrium-based binding assays with reference subtraction. (**F**) Luciferase

*Figure 6. continued on next page*

*Figure 6. Continued*

reporter assays histograms with *Kd* values from SPR measurements (*Figure 6—figure supplement 1*) shown above. Residues involved in the Fz4$_{CRD}$ binding site are coloured red. Residues without contact with Fz4$_{CRD}$ are coloured black. Grey filled boxes highlight disease-associated residues (*Figure 2—figure supplement 2*). The luciferase activities were normalized to a maximum activity value (100%) for Norrin wild-type and error bars represent standard deviations (n = 3).

The following figure supplement is available for figure 6:

**Figure supplement 1**. SPR equilibrium binding data.

selective and high affinity binding relative to interactions with other CRD of Fz receptors, in agreement with prior studies (*Xu et al., 2004*; *Smallwood et al., 2007*; *Ke et al., 2013*). However, it remains to be clarified whether the low affinity interactions of Norrin with other Fz receptors can play any functional role in vivo.

## Verification of GAG binding site

To assess our putative binding site for GAGs (*Figure 5E*), we performed structure-guided mutagenesis and functional studies. Our heparin binding experiments confirmed that Norrin shows high affinity interaction with heparin (*Figure 7—figure supplement 1A*), consistent with previous studies (*Perez-Vilar and Hill, 1997*; *Xu et al., 2004*; *Smallwood et al., 2007*; *Ohlmann et al., 2010*), and further demonstrated Norrin–Fz4$_{CRD}$ complex binding to heparin (*Figure 7A*). The Norrin triple mutation R107E/R109E/R115L (R115L is a disease-associated mutation; *Figure 2—figure supplement 2*) impaired heparin binding (*Figure 7B*) and abolished signalling activity (*Figure 7C*). However, this mutant protein retained the ability to bind Fz4$_{CRD}$ (*Figure 7D*) with a 2:2 stoichiometry (*Figure 7—figure supplement 1B*). *Ke et al. (2013)* have reported MBP-Norrin binding to the Lrp6 ectodomain fragment comprising the first two tandem β-propeller-epidermal growth factor-like domain pairs (Lrp6$_{P1E1P2E2}$; *Ke et al., 2013*); we found both our wild-type and R107E/R109E/R115L mutant Norrin bind to Lrp6$_{P1E1P2E2}$ (*Figure 7E,F*). The Norrin K58N mutant (a disease-associated mutation; *Figure 2—figure supplement 2*) exhibited half of wild-type activity in our cell-based assay (*Figure 7C*), but did not affect Fz4$_{CRD}$ interaction (*Figure 6—figure supplement 1C*). These results are in agreement with previous functional studies (*Smallwood et al., 2007*), and suggest this area is a GAG binding site rather than that, as *Ke et al. (2013)* proposed, residues Arg107, Arg109, and Arg115 are involved in Lrp5/6 binding (*Ke et al., 2013*). HSPGs play important roles in the regulation of the Wnt signalling pathway (*Malinauskas and Jones, 2014*). Wnt signalling activity can be inhibited by treatment with exogenous heparin (*Ai et al., 2003*). Also, *Jung et al. (2015)* have reported that PG545, a heparan sulphate mimetic, can block Wnt binding to the cell surface, by competing with endogenous HSPGs, and inhibit Wnt signalling (*Jung et al., 2015*). For Norrin mediated Wnt/β-catenin signalling, we found that SOS could inhibit activity when pre-incubated with Norrin before stimulation of reporter cells (*Figure 7—figure supplement 1D*).

## Mapping a potential Lrp5/6 binding site on Norrin

Norrin interaction with co-receptor Lrp5/6$_{ECD}$ (*Figure 7E*) is essential for signal activation (*Xu et al., 2004*; *Ke et al., 2013*). To identify Norrin residues potentially involved in Lrp5/6$_{ECD}$ binding, we assessed solvent exposure, disease-association, and lack of involvement in Fz4$_{CRD}$ or GAG binding (*Figure 2—figure supplement 2*). Five residues (Lys54, Arg90, Arg97, Gly112, and Arg121) were highlighted by this analysis and form a continuous, positively charged, concave patch (*Figure 8A*). Notably, a negatively charged region of the Lrp6$_{ECD}$ surface has been implicated in ligand binding (*Ahn et al., 2011*; *Bourhis et al., 2011*; *Chen et al., 2011*; *Cheng et al., 2011*). We therefore focused on the positively charged concave surface of Norrin as a potential Lrp5/6 binding site (*Figure 8A*), interestingly, this putative binding site has a partially overlap, at Lys54, with the residue suggested to be involved in Lrp5/6 interaction by *Ke et al. (2013)*. To test our proposed location for the Lrp5/6 binding site, we generated the disease-associated Norrin mutant R121W (Arg121 is a mutational hotspot; *Figure 2—figure supplement 2*). This mutation substantially impairs signalling activity (*Figure 8—figure supplement 1A*), but retains the ability to interact with Fz4$_{CRD}$ (*Figure 8—figure supplement 1B*) and heparin (*Figure 8—figure supplement 1C*). However, we found the R121W mutation reduced protein solubility and stability during protein

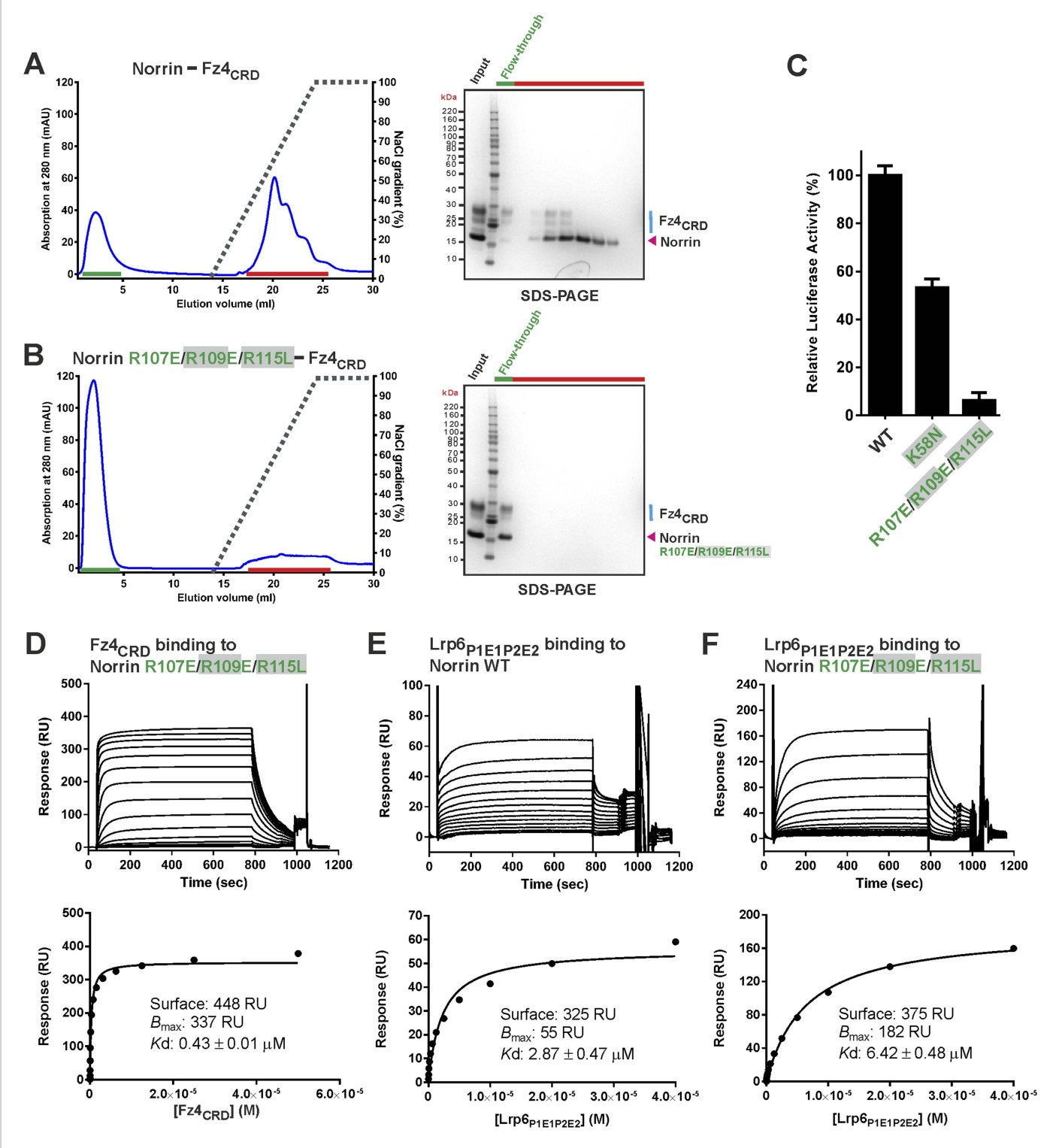

**Figure 7**. Verification of Norrin GAG binding site. Heparin affinity chromatography of (**A**) Norrin–Fz4$_{CRD}$ complex and (**B**) Norrin R107E/R109E/R115L–Fz4$_{CRD}$ complex. Protein elution profiles (left panel) were monitored by absorbance at 280 nm (blue curves) for a NaCl gradient (0.25–2 M; black dashed lines). Input sample, flow-through (green line) and peak fractions (red line) were analysed on SDS-PAGE (right panel). Norrin-Fz4$_{CRD}$ complex was eluted at 1.3 M NaCl concentration. (**C**) Luciferase reporter assays for Norrin mutations (coloured green) in the GAG binding site. Grey filled boxes highlight disease-associated residues (**Figure 2—figure supplement 2**). (**D**) SPR binding assay of Norrin R107E/R109E/R115L mutant and Fz4$_{CRD}$ interaction. Sensorgrams (top panel) and fitted plots of equilibrium binding response (bottom panels) for a series of concentrations of Fz4$_{CRD}$ are shown.

*Figure 7. continued on next page*

Figure 7. Continued

(**E** and **F**) SPR equilibrium binding experiments of Lrp6$_{P1E1P2E2}$ binding to Norrin wild-type and R107E/R109E/R115L mutant, respectively. Biotinylated Norrin proteins were immobilized on a CM5 chip and Lrp6$_{P1E1P2E2}$ as analyte was injected over the chip. Sensorgrams (top panel) and fitted plots (bottom panels) for a series of concentrations of Lrp6$_{P1E1P2E2}$ are presented.

The following figure supplement is available for figure 7:

**Figure supplement 1**. Supporting experiments for GAG binding site.

production and in heparin binding assays. Analyses of additional Norrin mutants in biophysical and cellular assays will be required to verify the putative Lrp5/6 binding site. Taken together, the current data suggest three distinct and independent binding sites on Norrin for Fz4, Lrp5/6, and GAGs (*Figure 8B*). This arrangement of binding sites likely enables Norrin to form a ternary complex.

## Structural comparison of Norrin–Fz4$_{CRD}$ with Wnt8–Fz8$_{CRD}$

As Norrin and Wnt both trigger the canonical Wnt/β-catenin pathway, we compared their modes of action. *Xenopus* Wnt8 (*Figure 9A*) has been described as using 'thumb' and 'index finger' regions to grasp mouse Fz8$_{CRD}$ at two distinct sites (*Janda et al., 2012*). In site 1, a palmitoleoyl group (PAM) covalently linked to the tip of the thumb inserts into a groove in Fz8$_{CRD}$, removal of this PAM moiety suppresses Wnt signalling activity (*Kakugawa et al., 2015*; *Zhang et al., 2015*). In site 2, the index finger contacts a hydrophobic pocket. We superposed Norrin–Fz4$_{CRD}$ with Wnt8–Fz8$_{CRD}$. There are no major structural differences between the Fz4$_{CRD}$ and Fz8$_{CRD}$ (r.m.s. deviation of 1.3 Å over 110 equivalent Cα atoms; *Figure 9A*), and the structural elements that mediate site 1 PAM binding in Fz8$_{CRD}$ are largely conserved in Fz4$_{CRD}$ (*Figure 9B*). The Norrin binding site on Fz4$_{CRD}$ (~800 Å$^2$ buried area) overlaps with site 2 on Fz8$_{CRD}$ (~400 Å$^2$ buried area; *Figure 8A*), in agreement with previous mutational mapping studies (*Small-wood et al., 2007*). The position of the Wnt8 index finger overlaps with Norrin β1 and β2, and, unexpectedly, these β strands show some structural equivalence with Wnt8 (*Figure 9C*). Site 2 Wnt8 residues are strictly conserved in all Wnts, and the apolar residues in the corresponding positions on Norrin are associated with disease mutations (*Figure 9C*).

We also used our superposition of the Fz4$_{CRD}$ and Fz8$_{CRD}$ structures (*Figure 9A*) to identify the determinants of the Norrin binding specificity for Fz4$_{CRD}$ (*Figure 6—figure supplement 1F*). In Fz4$_{CRD}$ loop I (*Figure 9D*), Asn55 is replaced by Fz8$_{CRD}$ Gly45, a change that would abolish interaction with Norrin Ser34 in the complex (*Figure 5B*). In Fz4$_{CRD}$ loop II (*Figure 9E*), the substitution of Lys109 by Fz8$_{CRD}$ Asp99 would introduce an unfavorable electrostatic interaction with Norrin Asp46 (*Figure 5C*). Thirdly, in Fz4$_{CRD}$ loop III (*Figure 9F*), hydrogen bonds and salt

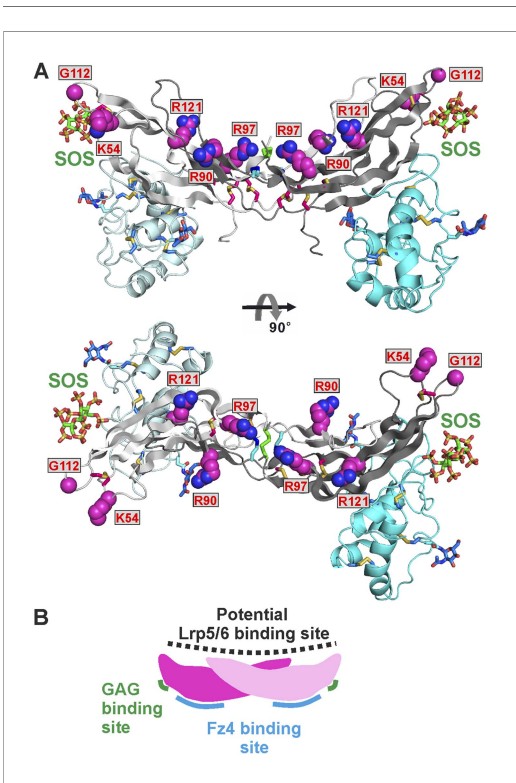

**Figure 8**. The potential Lrp5/6 binding site on Norrin. (**A**) Cartoon representation of Norrin (grey) in complex with Fz4$_{CRD}$ (cyan). Residues in the potential Lrp5/6 binding site are shown as spheres (atom colouring: magenta, carbon; blue, nitrogen; red, oxygen). The boxes highlight residues associated with disease mutations. (**B**) Cartoon model of Norrin showing three distinct binding sites.

The following figure supplement is available for figure 8:

**Figure supplement 1**. Verification of Norrin potential Lrp5/6 binding site.

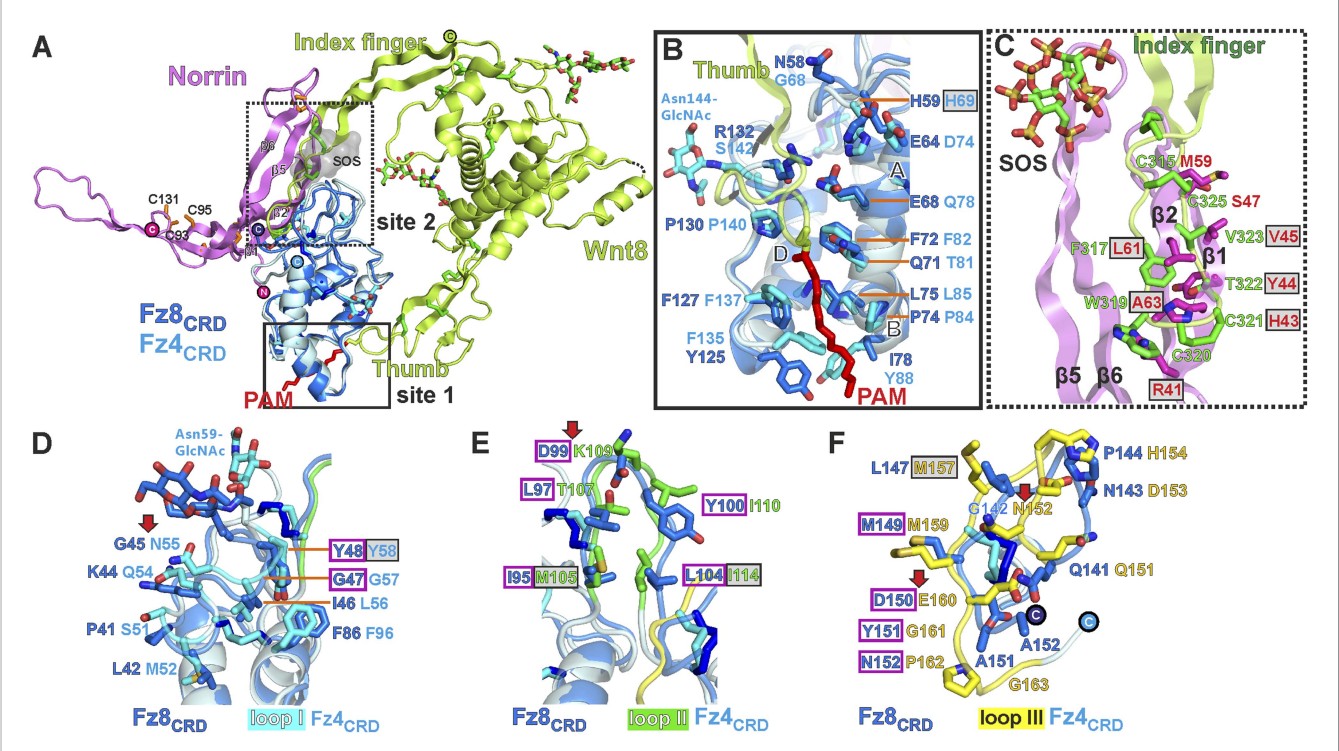

**Figure 9**. Structural comparison of Norrin–Fz4$_{CRD}$ with Wnt8–Fz8$_{CRD}$. (**A**) Superposition of Norrin (magenta)–Fz4$_{CRD}$ (cyan) with Wnt8 (green)–Fz8$_{CRD}$ (blue) (PDB ID: 4F0A). Disulphide bonds, N-linked glycans, and PAM (of Wnt8) are shown as sticks. SOS is shown as grey surface. (**B**) Comparison of site 1 (PAM binding) on Fz4$_{CRD}$ (cyan) and Fz8$_{CRD}$ (blue). Fz4 His69 is disease associated. (**C**) The Wnt8 index finger (site 2; green) structurally overlays Norrin (β1 and β2; magenta). Norrin residues associated with diseases are boxed. (**D–F**), Structural comparison of Fz4$_{CRD}$ and Fz8$_{CRD}$ for ligand binding. Loop I-III residues for Fz4$_{CRD}$ and Fz8$_{CRD}$ are shown as sticks. Fz8$_{CRD}$ residues for Wnt8 binding (site 2) are boxed in purple. Fz4 disease-associated residues are boxed. Red arrows indicate residue substitutions between Fz4$_{CRD}$ and Fz8$_{CRD}$. Fz8$_{CRD}$ residues Tyr151 and Asn152 are modelled as alanines (PDB ID: 4F0A).

bridges to Norrin would be lost on replacing Asn152 and Glu160 with Fz8$_{CRD}$ Gly142 and Asp150 respectively (*Figure 5D*). Consistent with this analysis, these residue substitutions have been reported to affect Fz4$_{CRD}$ binding to Norrin (*Smallwood et al., 2007*), and Fz4$_{CRD}$ is unique in containing this particular combination of residues (*Figure 2—figure supplement 2*).

## Discussion

Overall, our analyses provide several advances for our understanding of Norrin and Wnt signalling. Firstly, our results give fresh insight into the role of HSPGs. HSPGs have been proposed to regulate the local distribution of ligand and receptor at the cell surface, potentially acting as an introductory agency for ligand and receptor (*Lin and Perrimon, 1999*; *Baeg et al., 2001*; *Malinauskas et al., 2011*; *Malinauskas and Jones, 2014*). We have discovered a GAG binding site that may span Norrin and Fz4$_{CRD}$ (*Figure 5E*). Interestingly, *Smallwood et al. (2007)* found the binding affinity of Norrin with Fz4$_{CRD}$ is enhanced in the presence of heparin (*Smallwood et al., 2007*). We propose that the extended GAG binding site may allow co-receptor HSPGs to recruit secreted Norrin for interaction with Fz4$_{CRD}$ and to co-localize Norrin and Fz4 receptor, similar to the role of HSPGs in Wnt signalling (*Reichsman et al., 1996*; *Baeg et al., 2001*; *Fuerer et al., 2010*). For example, HSPGs have been shown to regulate the Wnt morphogenetic gradient (*Lin and Perrimon, 1999*; *Baeg et al., 2004*). Also, *Capurro et al. (2014)* have reported that Fz4$_{CRD}$ binds to the GAGs of the human HSPG Glypican-3 and that these interactions are involved in Wnt signal complex formation (*Capurro et al., 2014*). Secondly, we show the Norrin dimer binds separately to two molecules of Fz4$_{CRD}$ (*Figure 4*), in contrast to the 1:1 complex of Wnt8–Fz8$_{CRD}$ (*Janda et al., 2012*). Our discovery of the Fz4 and GAG

binding sites, and analysis of a potential Lrp5/6 binding region, maps out distinct binding surfaces on Norrin (*Figure 8B*), which provide a framework in which to understand the effects of inherited mutations and probe the overall architecture of the ternary complex (Norrin–Fz4$_{CRD}$–Lrp5/6$_{ECD}$). Thirdly, we determine how Norrin structurally mimics Wnt for site 2 binding surfaces on the Fz ectodomain (*Figure 9A*). Interestingly, previous analyses using water-soluble 'mini-Wnt' proteins, which cannot contribute site 1 binding, have raised the possibility that site 2 binding to the CRD of Fz receptors can activate canonical Wnt/β-catenin signalling albeit weakly (*Janda et al., 2012*; *von Maltzahn et al., 2013*). Our findings indicate that the site 2 binding mode is central to signallosome formation for Norrin mediated signalling.

We used SPR experiments to establish the binding affinity for Norrin–Fz4$_{CRD}$ complex formation. The Kd value of 1.1 μM for the interaction between Norrin and Fz4$_{CRD}$ we report here (*Figure 6—figure supplement 1A*) is weaker than previously published results (*Xu et al., 2004*; *Ke et al., 2013*). This discrepancy is likely due to our SPR binding assays being carried out with monomeric Fz4$_{CRD}$. *Xu et al. (2004)* used an enzyme-linked immunosorbent assay to give an affinity of 3–4 nM for mouse Norrin fused with C-terminal alkaline phosphatase binding to mouse Fz4$_{CRD}$ dimerized by a C-terminal Fc fusion (*Xu et al., 2004*). *Ke et al. (2013)* reported Kd values of 11 nM and 5 nM for the interaction between MBP-Norrin and Fc-tagged dimeric Fz4$_{CRD}$ using an AlphaScreen luminescence assay and biolayer interferometry, respectively (*Ke et al., 2013*). It is noteworthy that as Fc-dimerized Fz4$_{CRD}$ may mimic Fz4 receptor dimerization at the cellular surface, these tighter binding affinities may be more indicative of Norrin binding in the physiologically relevant environment. Similarly, our Kd value of 2.87 μM for Norrin binding to Lrp6$_{P1E1P2E2}$ in an SPR based assay (*Figure 7E*) differs from the Kd value of 0.45 μM reported by *Ke et al. (2013)* based on an homologous AlphaScreen competition assay using unlabeled MBP-Norrin against biotinylated MBP-Norrin for interaction with Lrp6$_{P1E1P2E2}$ (*Ke et al., 2013*).

Our studies reported here, in combination with previous findings for Norrin and Wnt signalling, are consistent with Norrin-induced receptor clustering and signallosome formation. Inactive pre-dimerized Fz4 may engage with homodimeric Tspan-12 to enhance receptor clustering (*Kaykas et al., 2004*; *Ke et al., 2013*). Norrin binding generates ternary complex formation by Fz4, Lrp5/6 and the GAGs of HSPGs to trigger signalling, which is enhanced in the presence of Tspan12. In the cytoplasm, Dishevelled binds to the C-terminal tail of Fz4 and self-assembles to oligomer (*Schwarz-Romond et al., 2007*), leading to Axin recruitment to the cytoplasmic domain of Lrp5/6 for phosphorylation and signallosome formation (*Bilic et al., 2007*).

Previously reported mice genetic studies have demonstrated that expression of ectopic Norrin can rescue pathological retinal vascularization (*Ohlmann et al., 2005*, *2010*). In addition, the pathological progresses of Norrie disease and familial exudative vitreoretinopathy are highly related to age-related macular degeneration and diabetic retinopathy (*Ye et al., 2010*; *Ohlmann and Tamm, 2012*). Further investigation of the therapeutic possibilities for retinal diseases has been hampered by the difficulty of producing recombinant Norrin proteins. In this study, we provide a method to produce fully active untagged Norrin in mammalian cells (*Figure 1*). Our recombinant Norrin opens up new avenues to explore for the treatment of genetic retinal diseases and other ophthalmic disorders.

More generally, Norrin as a Wnt mimic, may have potential as a reagent in regenerative medicine (*Clevers et al., 2014*).

Wnt signalling is important for tissue homeostasis throughout life (*Clevers and Nusse, 2012*). Multiple Wnt extracellular antagonists function to modulate Wnt signalling (*Malinauskas and Jones, 2014*), these include Dickkopf and Sclerostin, which bind to Lrp5/6, as well as Wnt inhibitory 1 and sFRPs, which sequester Wnt. Aberrant Wnt signalling (insufficient or excessive) is implicated in diseases such as neurodegeneration and tumorigenesis, respectively (*Clevers and Nusse, 2012*; *Anastas and Moon, 2013*). Interestingly, our Norrin mutants (used to verify the GAG and putative Lrp5/6 binding sites) retain Fz4$_{CRD}$ binding but lose the ability to activate signalling. These properties are similar to those of the monoclonal antibody OMP-18R5 which can bind to the CRDs of Fz1, 2, 5, 7 and 8. OMP-18R5 inhibits tumour growth (*Gurney et al., 2012*) and has just completed phase I clinical trials (*Kahn, 2014*). Engineered Norrin mutants could similarly serve as blocking agents, but with specificities tailored to target Fz4 or other individual Fz receptors.

## Materials and methods

### Construct design, cloning, and mutagenesis

Synthetic complementary DNA (cDNA) clones (codon-optimized for expression in mammalian cells) of human Norrin (UniprotKB/Swiss-prot Q00604) were obtained from GeneArt (Life Technologies, UK). The cDNA templates of human receptors Fz4 (IMAGE ID: 40082087), Fz7 (IMAGE ID: 4549389), Lrp6 (IMAGE ID: 40125687), and Tspan-12 (IMAGE ID: 5275953) and mouse receptors Fz5 (IMAGE ID: 40088671) and Fz8 (IMAGE ID: 8861081) were purchased from SourceBioScience (UK). All expression constructs reported here are derived from the pHLsec vector backbone (*Aricescu et al., 2006*). The human Norrin wild-type (residues 25–133) construct of SUMO-Norrin (*Figure 1A*) and Norrin mutant constructs for heparin affinity binding assays were tagged N-terminally with the murine Igκ-chain secretion signal, followed by a Strep-II tag, 8xHis tag, a mammalian expression codon-optimized *Saccharomyces cerevisiae* SUMO (UniprotKB/Swiss-prot Q12306; residues 2–96) (*Peroutka et al., 2008*), and a Human Rhinovirus (HRV)-3C protease cleavage site. They were tagged C-terminally with a TETSQVAPA sequence derived from bovine rhodopsin (Rho-1D4) that is recognized by the Rho-1D4 monoclonal antibody (*Molday and MacKenzie, 1983*). The construct of Norrin (residues 25–133) was cloned into the pHLsec vector (*Aricescu et al., 2006*) in frame with a C-terminal Rho-1D4 (*Figure 1A*). For large-scale protein expression, the CRD constructs for human Fz4 (residues 42–179) and Fz7 (residues 42–179) as well as mouse Fz5 (residues 31–176) and Fz8 (residues 30–170) were cloned into a modified pHLsec vector (pHLsec-mVenus-12H), containing a C-terminal HRV-3C protease cleavage site followed by a linker, monoVenus (*Nagai et al., 2002*), and a tandem 6×His tag. Human Lrp6$_{P1E1P2E2}$ (residues 1–631) construct was cloned into a modified vector for stable cell line generation, pNeoSec (*Zhao et al., 2014*), in frame with a C-terminal 10×His tag. For luciferase reporter assays, the full-length constructs of the human receptors Fz4 (residues 1–537), Lrp6 (residues 1–1613), and Tspan-12 (residues 1–305) were cloned into the pLEXm-1D4 vector carrying a C-terminal Rho-1D4 tag. Norrin wild-type and mutants for biophysical and cellular assays were obtained from GeneArt (Life Technologies, UK) and cloned into the pHL-Avitag3 vector encoding a C-terminal BirA recognition sequence (*Aricescu et al., 2006*). Mutant proteins were secreted at similar levels to the wild-type proteins. Constructs were verified by DNA sequencing (Source Bioscience, UK).

### Western blot assays

For western blot, HEK293T (ATCC CRL-11268) cells were transfected with the DNA using Lipofectamine 2000 (Life Technologies, UK) according to the manufacturer's instructions. The conditioned media were collected 2 days post transfection and were analysed by sodium dodecyl sulfate polyacrylamide gel electrophoresis (SDS-PAGE) gels transferred onto a nitrocellulose membrane (GE Healthcare Life Sciences) with Rho-1D4 monoclonal antibodies (Flintbox, University of British Columbia, Canada) as primary antibody and goat anti-mouse IgG-horseradish peroxidise conjugate (Sigma). The signal was visualized by Enhanced Chemiluminescence western blotting detection kit (ECL, GE Healthcare Life Sciences).

### Protein production and purification

Norrin wild-type and mutants were expressed in HEK293T cells (*Aricescu et al., 2006*) in the presence of 4 mM valproic acid (*Backliwal et al., 2008*). For crystallization experiments, Fz4$_{CRD}$ was produced in HEK293T cells in the presence of 5 μM of the class I α-mannosidase inhibitor, kifunensine (*Chang et al., 2007*). Norrin in complex with Fz4$_{CRD}$ was co-expressed in HEK293T cells in the presence of kifunensine and valproic acid. For all other experiments, recombinant proteins were expressed in HEK293T cells. The Norrin conditioned media were passed through 1D4-affinity beads covalently coupling purified Rho-1D4 antibody to CnBr-activated Sepharose 4 Fast Flow (CnBr-1D4; GE Healthcare Life Sciences) and eluted in 25 mM Tris, pH 7.5, 0.5 M NaCl, 10% (wt/vol) Glycerol, 0.5% (wt/vol) CHAPS, 250 μM TETSQVAPA peptide (GenScript). The eluted sample was incubated with Glutathione S-Transferase (GST)-tagged HRV-3C protease to remove the SUMO-tagged fusion protein. The cleaved Norrin was purified by CnBr-1D4 followed by SEC (Superdex 200 10/300 GL High Performance, GE Healthcare Life Sciences) in either 10 mM HEPES, pH 7.5, 0.7 M NaCl, 0.5% (wt/vol) CHAPS or acetate buffer, pH 4.0, 0.5 M NaCl, 0.5% (wt/vol) CHAPS. For purification of Fz4$_{CRD}$, the

conditioned media were dialyzed and recombinant proteins were purified by IMAC (TALON beads, Clontech, Mountain View, CA). The purified sample was dialyzed against 25 mM Tris, pH 7.5, 0.5 M NaCl, 10% (wt/vol) Glycerol and treated with GST-tagged *Flavobacterium meningosepticum* endoglycosidase-$F_1$ (Endo-$F_1$) (*Chang et al., 2007*) and His-tagged HRV 3C protease. The deglycosylated and cleaved sample was further purified by IMAC and further polished by SEC (Superdex 75 16/600 column, GE Healthcare Life Sciences) in 10 mM HEPES, pH 7.5, 0.15 M NaCl. Purification of $Fz5_{CRD}$, $Fz7_{CRD}$, and $Fz8_{CRD}$ followed the same procedure to that described above, except protein was expressed in HEK293T cells and the treatment by Endo-$F_1$ was omitted. Norrin–$Fz4_{CRD}$ complex was isolated from dialyzed conditioned media by IMAC. The eluted sample was dialyzed and treated with GST-tagged HRV-3C protease and Endo-$F_1$. The deglycosylated and cleaved complex was further purified by IMAC and GST-affinity beads and subsequently isolated by SEC (Superdex 200 16/600 column, GE Healthcare Life Sciences) in 10 mM HEPES, pH 7.5, 0.7 M NaCl. For preparation of methylated proteins, the purified sample was subject to surface lysine methylation (*Walter et al., 2006*) and further purified by SEC (Superdex 200 16/600 column, GE Healthcare Life Sciences). The selenomethionine (Se-Met) labelled protein was prepared as described previously (*Aricescu et al., 2006*). A stable HEK293 GnT1(−) cell line (*Reeves et al., 2002*) for $Lrp6_{P1E1P2E2}$ protein production was generated as reported previously (*Zhao et al., 2014*) and protein was purified following our established procedure (*Chen et al., 2011*).

## Crystallization and data collection

Concentrated proteins (Norrin, 5 mg/ml; $Fz4_{CRD}$, 60 mg/ml; Norrin in complex with $Fz4_{CRD}$ including native and methylated proteins, 10–12 mg/ml) were subjected to sitting drop vapor diffusion crystallization trials in 96-well Greiner plates consisting of 100 nl protein solution and 100 nl reservoir using a Cartesian Technologies dispensing instrument (*Walter et al., 2005*). Crystallization plates were placed in a The Automation Partnership storage vault maintained at 294 K and imaged via a Veeco visualization system. Methylated Norrin–$Fz4_{CRD}$ complex crystallized in 0.1 M Bicine, pH 9.0, 10% (wt/vol) PEG6000, Norrin crystal form I in 0.1 M sodium acetate, pH 5.0, 5% (wt/vol) PGA-LM, 30% (wt/vol) PEG550MME, Norrin crystal form II in 0.1 sodium acetate, pH 5.0, 5% (wt/vol) PGA-LM, 4% (wt/vol) PEG2000MME, 24% (wt/vol) PEG550MME, Norrin crystal form III in 0.1M citrate, pH 5.0, 30% (wt/vol) PEG6000, $Fz4_{CRD}$ crystal form I in 1.6 M tri-sodium citrate, pH 6.5, and $Fz4_{CRD}$ crystal form II in 0.1 M HEPES, pH 7.5, 0.1 M NaCl, 1.6 M ammonium sulfate. For the Norrin–$Fz4_{CRD}$–SOS complex, protein complex was mixed with 10 mM SOS (Toronto Research Chemicals Inc.) prior to crystallization and crystals were obtained in 0.1 M Tris, pH 8.0, 0.15 M NaCl, 8% (wt/vol) PEG8000. For cryoprotection, crystals were soaked in mother liquor supplemented with 30% (vol/vol) glycerol for methylated Norrin–$Fz4_{CRD}$, with 20% (vol/vol) PEG200 and 10 mM SOS for Norrin–$Fz4_{CRD}$–SOS, with 30% (vol/vol) PEG550MME for Norrin crystal form II, with 30% (vol/vol) glycerol for Norrin crystal form III, with 1.8 M tri-sodium citrate, pH 6.5 for $Fz4_{CRD}$ crystal form I, and with 23% (vol/vol) sucrose for $Fz4_{CRD}$ crystal form II and subsequently flash-cooled by dipping into liquid nitrogen. The crystals of Norrin crystal form I were frozen directly. Data were collected at 100 K at Diamond Light Source (Oxfordshire, UK) at beamlines I03 (Norrin Se-Met), I04 (methylated Norrin–$Fz4_{CRD}$ and Norrin crystal form II and III), I04-1 (Norrin–$Fz4_{CRD}$–SOS), and I24 (Norrin crystal form I and $Fz4_{CRD}$ crystal form I and II). Diffraction data were indexed and integrated using XIA2 (*Winter, 2010*) coupled with XDS or IMOSFLM, and scaled and merged using Aimless (*Evans and Murshudov, 2013*). A subset of 5% of randomly selected diffraction data were used for calculating $R_{free}$ (*Brunger, 1993*).

## Structure determination and refinement

The structure of Norrin crystal form I was solved using highly redundant single-wavelength anomalous dispersion data merged from four data sets and collected at the Se K absorption edge. HKL2MAP (*Sheldrick, 2010*) was used to identify the Se sites, which were then fed into PHENIX AUTOSOL (*Adams et al., 2002*), resulting in an interpretable density modified electron map generated by RESOLVE (*Terwilliger, 2003*). An initial model generated by BUCCANEER (*Cowtan, 2006*) was used to solve the high-resolution native structures. The structure of $Fz4_{CRD}$ was determined by molecular replacement (MR) in PHASER (*McCoy, 2007*) using mouse $Fz8_{CRD}$ (PDB ID: 1IJY) as the search model, which was modified by CHAINSAW. For the determination of methylated Norrin–$Fz4_{CRD}$, Norrin was

used as search model for MR in PHASER (*McCoy, 2007*) to obtain the initial phases. The additional electron density corresponding to Fz4$_{CRD}$ was clearly discernible after density modification with PARROT (*Cowtan, 2010*). Subsequently, the complex structure was solved by searching for Fz4$_{CRD}$ with MR in PHASER (*McCoy, 2007*). All other structures were solved by MR in PHASER (*McCoy, 2007*) using the refined Norrin and Fz4$_{CRD}$ structures as search models. The models were completed by manual rebuilding in COOT (*Emsley and Cowtan, 2004*) and refinement in REFMAC5 (*Murshudov et al., 1997*) and PHENIX (*Adams et al., 2010*). The crystallographic statistics are listed in *Table 1*. All models were validated with MOLPROBITY (*Chen et al., 2010*).

## Structure analysis

Amino acid sequence alignments were constructed using ClustalW (*Thompson et al., 1994*). Structure superposition was performed within the CCP4 program suite using the SSM algorithm (*Krissinel and Henrick, 2004*). Electrostatic potential calculations were generated using APBS tools (*Baker et al., 2001*), surface sequence conservation was calculated using CONSURF (*Ashkenazy et al., 2010*) and interface areas of proteins were analyzed with the PISA web server (*Krissinel and Henrick, 2007*). High-quality images of the molecular structures were created with the PyMOL Molecular Graphics System (Version 1.5, Schrödinger, LLC). Schematic figures and other illustrations were prepared using Corel Draw (Corel Corporation).

## Surface plasmon resonance equilibrium binding studies

SPR experiments were performed using a Biacore T200 machine (GE Healthcare Life Sciences) at 25°C in 10 mM HEPES, pH 7.5, 0.15 M NaCl, 0.005% (wt/vol) Tween20. For in vivo biotinylation (*Penalva and Keene, 2004*), Norrin wild-type or mutants in the pHL-Avitag3 vector (*Aricescu et al., 2006*) were co-transfected with a pHLsec construct of BirA-ER (the synthetic BirA gene with a C-terminal KDEL sequence for retention in the endoplasmic reticulum) in HEK293T cells. Mutant proteins were secreted at similar levels to the wild-type proteins. The mammalian cell secretory pathway uses stringent quality control mechanisms to ensure that secreted proteins are correctly folded (*Trombetta and Parodi, 2003*). The biotinylated Norrin variants were immobilized onto the surface of a CM5 sensor chip (GE Healthcare Life Sciences) on which approximately 8500 resonance units of streptavidin were coupled via primary amines. Fz4$_{CRD}$ proteins used as analytes were expressed in HEK293T cells to ensure full glycosylation and prepared as described above. The signal from SPR flow cells was corrected by subtraction of a blank and reference signal from a mock-coupled flow cell. In all analyses, the experimental trace returned to baseline line after a regeneration step with 100 mM phosphate pH 3.7, 2 M NaCl, 1% (wt/vol) Tween20. The data were fitted to a 1:1 Langmuir adsorption model ($B = B_{max}C/(K_d + C)$, where $B$ is the amount of bound analyte and $C$ is the concentration of analyte in the sample) for the calculation of dissociation constant ($K_d$) values using Biacore Evaluation software (GE Healthcare Life Sciences). Data points correspond to the average from two independent dilution series.

## Small-angle x-ray scattering experiment

Solution scattering data were collected at beamline BM29 of the European Synchrotron Radiation Facility (ESRF; Grenoble, France) at 293 K within a momentum transfer range of 0.01 Å$^{-1}$ < $q$ < 0.45 Å$^{-1}$, where $q = 4\pi\sin(\theta)/\lambda$ and $2\theta$ is the scattering angle (*Pernot et al., 2013*). X-ray wavelength was 0.995 Å and data were collected on a Pilatus 1M detector. Fz4$_{CRD}$ was measured at 1.47 and 3.10 mg/ml (deglycosylated form) and 0.97 and 1.45 mg/ml (glycosylated form) in 10 mM HEPES pH 7.5, 0.15 M NaCl. Norrin was measured at 0.75 and 1.26 mg/ml in 10 mM HEPES, pH 7.5, 0.7 M NaCl, 0.5% (wt/vol) CHAPS. The deglycosylated Norrin–Fz4$_{CRD}$ complex was measured at 1.02 and 2.14 mg/ml in 10 mM HEPES, 0.5 M NaCl. Data reduction and calculation of invariants was carried out using standard protocols implemented in the ATSAS software suite (*Petoukhov et al., 2012*). A merged dataset was obtained by merging the low-angle part of the low-concentration dataset with the high-angle part of the high-concentration dataset. The Radius of gyration ($R_g$) was obtained from Guinier plot using AutoRg (*Petoukhov et al., 2012*). The maximum dimension of the particle ($D_{max}$) and Volume Porod ($V_p$ [nm$^3$]) were calculated by GNOM (*Svergun, 1992*). Molecular weights were obtained by (a) comparison with the reference bovine serum albumin (BSA) and (b) dividing the Porod Volume by 1.66 (*Rambo and Tainer, 2011*). Theoretical X-ray scattering patterns of structural models were calculated and fitted to experimental X-ray scattering curves using the program FoXS (*Schneidman-Duhovny et al., 2010*).

The Norrin, Fz4$_{CRD}$ and the Norrin–Fz4$_{CRD}$ complex solution structures were modeled starting from their respective crystal structures. Complex glycan structures and missing regions of N- and C-termini were added using the program Modeller (*Eswar et al., 2003*). All-atom simulations, and calculation and fitting of scattering patterns of Norrin, Fz4$_{CRD}$ and the Norrin–Fz4$_{CRD}$ complex were performed using the automated AllosMod-FoXS procedure (*Guttman et al., 2013*).

## Size-exclusion chromatography coupled to multi-angle light scattering analysis

SEC-MALS experiments were performed by using SEC on an analytical Superdex S200 10/300 GL column (GE Healthcare Life Sciences) connected to online static light-scattering (DAWN HELEOS II, Wyatt Technology, Santa Barbara, CA), differential refractive index (Optilab rEX, Wyatt Technology, Santa Barbara, CA) and Agilent 1200 UV (Agilent Technologies, Santa Clara, CA) detectors. Purified sample (Fz$_{CRD}$ proteins at 50 µM or Norrin–Fz4$_{CRD}$ complex at 25 µM) was injected into a column equilibrated in 10 mM HEPES, pH 7.5, 0.15 mM NaCl. Molecular mass determination was performed using an adapted RI increment value (*dn/dc* standard value; 0.186 ml/g) to account for the glycosylation state. The theoretical molecular weight was predicated from amino acid sequence plus 2.35 kDa per N-linked glycosylation site for full glycosylated protein produced from HEK293T cells or 203 Da per site for deglycosylated protein produced from HEK293T cells in the presence of kifunensine (*Chang et al., 2007*) with limited glycosylation and treated with Endo-F$_1$. Data were analyzed using the ASTRA software package (Wyatt Technology, Santa Barbara, CA).

## Luciferase reporter assay

The stable HEK293STF cell lines (*Xu et al., 2004*) carrying the Super Top Flash firefly luciferase reporter were split into 96-well plates and transfected 24 hr later with 200 ng DNA per well using Lipofectamine 2000 (Life Technologies, UK) according to the manufacturer's instructions. For assessment of interface mutants used in SPR experiments, the DNA mix contained 80 ng Norrin plasmid, 40 ng each of Fz4 and Lrp6 plasmids, 20 ng each of Tspan-12 and constitutive *Renilla* luciferase plasmids (pRL-TK, Promega, Madison, WI). The firefly and *Renilla* luciferase activities were measured 48 hr later with Dual-Glo luciferase reporter assay system (Promega, Madison, WI) using an Ascent Lunimoskan luminometer (Labsystems). For evaluation of recombinant Norrin and SOS inhibition, the DNA mix (80 ng pLEXm plasmid, 40 ng each of Fz4 and Lrp6 plasmids, 20 ng each of Tspan-12 and pRL-TK plasmids) was used for transfection. Cells were stimulated 6 hr post transfection with 9 µg/ml Norrin, 9 µg/ml Norrin preincubated with 2 mM SOS for 15 min, or control 9 µg/ml Fetal Calf Serum (FCS). The Dual-Glo luciferase reporter assays were performed 48 hr later. The firefly luciferase activity was normalized to *Renilla* luciferase activity (relative light unit, RLU). Luciferase reporter assays were performed 3 times in triplicate.

## Heparin affinity chromatography

Protein samples produced in HEK293T cells were freshly purified by SEC and then adjusted in 50 mM Tris, pH 7.5, 0.25 M NaCl. Purified protein (0.5 mg) was loaded onto a 1 ml HiTrap heparin HP column (GE Healthcare Life Sciences) equilibrated in 20 mM Tris, pH 7.5, 0.25 M NaCl and eluted with a linear NaCl gradient to 20 mM Tris, pH 7.5, 2 M NaCl, 5% (wt/vol) glycerol over 10 column volumes. Notably, we found that Norrin–Fz4$_{CRD}$ complex tends to partially disassemble (Fz4$_{CRD}$ detected in flow-through; *Figure 7A*) during sample preparation for the heparin binding assay (NaCl concentration was reduced from 0.5M to 0.25M).

## Acknowledgements

We thank staff at Diamond Light Source (beamlines I03, I04, I04-1 and I24; proposal mx8423) and European Synchrotron Radiation Facility (beamline BM29); M Jones and T Walter for technical support; W Lu for tissue culture; Y Zhao for a Lrp6$_{P1E1P2E2}$ stable cell line; J Nathans for the gift of HEK293STF cells; G Schertler and J Standfuss for advice concerning the Rho-1D4 affinity purification method; A Clayton, N Mitakidis, and AR Aricescu for the gifts of pHLmVenus and pLEXm-1D4 vectors. This work was supported by grants to EYJ from Cancer Research UK (C375/A10976) and the UK Medical Research Council (G0900084). The WTCHG is supported by the Wellcome Trust (090532/Z/09/Z). THC was funded by a Nuffield Department of Medicine Prize Studentship in

conjunction with Clarendon and Somerville College Scholarships. MZ and JE were supported by Marie Curie IEF fellowships.

# Additional information

## Funding

| Funder | Grant reference | Author |
|---|---|---|
| Wellcome Trust | 090532/Z/09/Z | Tao-Hsin Chang, Fu-Lien Hsieh, Matthias Zebisch, Karl Harlos, Jonathan Elegheert, E Yvonne Jones |
| Cancer Research UK | C375/A10976 | Tao-Hsin Chang, Fu-Lien Hsieh, Matthias Zebisch, E Yvonne Jones |
| Medical Research Council | G0900084 | Karl Harlos, E Yvonne Jones |

The funders had no role in study design, data collection and interpretation, or the decision to submit the work for publication.

## Author contributions

T-HC, Development of a new mammalian expression system, Conception and design, Acquisition of data from protein production to structural and functional experiments, Analysis and interpretation of data, Drafting or revising the article; F-LH, Acquisition of data (tissue cultures, protein production, and functional assays), Analysis and interpretation of data; MZ, Acquisition of data (crystallography), Revising the article; KH, Acquisition of data (crystallography); JE, Acquisition of data (SEC-MALS and SAXS), Drafting SAXS section; EYJ, Conception and design, Drafting or revising the article

# Additional files

## Major datasets

The following datasets were generated:

| Author(s) | Year | Dataset title | Dataset ID and/or URL | Database, license, and accessibility information |
|---|---|---|---|---|
| Chang TH, Hsieh FL, Zebisch M, Harlos K, Jones EY | 2015 | Crystal structure of Norrin in complex with the cysteine-rich domain of Frizzled 4 and sucrose octasulfate | http://www.rcsb.org/pdb/explore/explore.do?structureId=5BQC | Publicly available at RCSB Protein Data Bank (Accession No. 5BQC). |
| Chang TH, Hsieh FL, Harlos K, Jones EY | 2015 | Crystal structure of Norrin in complex with the cysteine-rich domain of Frizzled 4 -Methylated form | http://www.rcsb.org/pdb/explore/explore.do?structureId=5BQE | Publicly available at RCSB Protein Data Bank (Accession No. 5BQE). |
| Chang TH, Hsieh FL, Harlos K, Jones EY | 2015 | Crystal structure of Norrin, a Wnt signalling activator, Crystal Form I | http://www.rcsb.org/pdb/explore/explore.do?structureId=5BPU | Publicly available at RCSB Protein Data Bank (Accession No. 5BPU). |
| Chang TH, Hsieh FL, Harlos K, Jones EY | 2015 | Crystal structure of Norrin, a Wnt signalling activator, Crystal Form II | http://www.rcsb.org/pdb/explore/explore.do?structureId=5BQ8 | Publicly available at RCSB Protein Data Bank (Accession No. 5BQ8). |
| Chang TH, Hsieh FL, Harlos K, Jones EY | 2015 | Crystal structure of Norrin, a Wnt signalling activator, Crystal Form III | http://www.rcsb.org/pdb/explore/explore.do?structureId=5BQB | Publicly available at RCSB Protein Data Bank (Accession No. 5BQB). |
| Chang TH, Hsieh FL, Harlos K, Jones EY | 2015 | Crystal structure of the cysteine-rich domain of human Frizzled 4 - Crystal Form I | http://www.rcsb.org/pdb/explore/explore.do?structureId=5BPB | Publicly available at RCSB Protein Data Bank (Accession No. 5BPB). |

| Author(s) | Year | Dataset title | Dataset ID and/or URL | Database, license, and accessibility information |
|---|---|---|---|---|
| Chang TH, Hsieh FL, Harlos K, Jones EY | 2015 | Crystal structure of the cysteine-rich domain of human Frizzled 4 - Crystal Form II | http://www.rcsb.org/pdb/explore/explore.do?structureId=5BPQ | Publicly available at RCSB Protein Data Bank (Accession No. 5BPQ). |

The following previously published datasets were used:

| Author(s) | Year | Dataset title | Dataset ID and/or URL | Database, license, and accessibility information |
|---|---|---|---|---|
| Dann III CE, Hsieh JC, Rattner A, Sharma D, Nathans J, Leahy DJ | 2001 | Crystal structure of the cysteine-rich domain of mouse Frizzled 8 (Mfz8) | http://www.rcsb.org/pdb/explore/explore.do?structureId=1IJY | Publicly available at RCSB Protein Data Bank (Accession No. 1IJY). |
| Ke J, Harikumar KG, Erice C, Chen C, Gu X, Wang L, Parker N, Cheng Z, Xu W, Williams BO, Melcher K, Miller LJ, Xu HE | 2013 | Crystal Structure of Norrin in fusion with Maltose Binding Protein | http://www.rcsb.org/pdb/explore/explore.do?structureId=4MY2 | Publicly available at RCSB Protein Data Bank (Accession No. 4MY2). |
| Janda CY, Waghray D, Levin AM, Thomas C, Garcia KC | 2012 | Crystal structure of XWnt8 in complex with the cysteine-rich domain of Frizzled 8 | http://www.rcsb.org/pdb/explore/explore.do?structureId=4F0A | Publicly available at RCSB Protein Data Bank (Accession No. 4F0A). |
| Dann III CE, Hsieh JC, Rattner A, Sharma D, Nathans J, Leahy DJ | 2001 | Crystal structure of the Cysteine-rich domain of secreted Frizzled-related protein 3 (Sfrp-3;Fzb) | http://www.rcsb.org/pdb/explore/explore.do?structureId=1IJX | Publicly available at RCSB Protein Data Bank (Accession No. 1IJX). |
| Stiegler AL, Burden SJ, Hubbard SR | 2009 | Crystal Structure of the Frizzled-like Cysteine-rich Domain of MuSK | http://www.rcsb.org/pdb/explore/explore.do?structureId=3HKL | Publicly available at RCSB Protein Data Bank (Accession No. 3HKL). |
| Nachtergaele S, Whalen DM, Mydock LK, Zhao Z, Malinauskas T, Krishnan K, Ingham PW, Covey DF, Siebold C, Rohatgi R | 2013 | Crystal structure of the Smoothened CRD, native | http://www.rcsb.org/pdb/explore/explore.do?structureId=4C79 | Publicly available at RCSB Protein Data Bank (Accession No. 4C79). |
| Rana R, Carroll CE, Lee HJ, Bao J, Marada S, Grace CR, Guibao CD, Ogden SK, Zheng JJ | 2013 | Solution structure of Smoothened | http://www.rcsb.org/pdb/explore/explore.do?structureId=2MAH | Publicly available at RCSB Protein Data Bank (Accession No. 2MAH). |

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
