## [Decision Letter]

Thank you for sending your work entitled “Structure and functional properties of Norrin mimic Wnt for signalling with Frizzled4, Lrp5/6, and proteoglycan” for consideration at *eLife*. Your article has been favorably evaluated by John Kuriyan (Senior editor) and two reviewers.

The Reviewing editor and the other reviewers discussed their comments before we reached this decision, and the editor has assembled the following comments to help you prepare a revised submission.

Chang et al. report structural and functional studies of Norrin, its binding partner Fz4_CRD_, and the Norrin:Fz4_CRD_ complex. They develop a mammalian expression system to produce functional Norrin, a member of a class of cystine-knot ligands that are often notoriously difficult to express. They crystallize and determine the structure of Norrin alone, of Fz4_CRD_ alone, and complexes of Norrin with Fz4_CRD_ and Fz4_CRD_ and sucrose-octasulfate (SOS), frequently used as a glycosaminoglycan (GAG) mimic. These structures, and the ensuing functional analysis, establish important features of Norrin biology. First, the authors show a 2:2 Norrin:Fz4_CRD_ binding stoichiometry, mediated by a large interface between each monomer within the Norrin dimer with a molecule of Fz4_CRD_ (located on opposite ends of the Norrin dimer). This interface is comprised of an extensive network of interacting residues on two beta-hairpins in Norrin with those on three loops in Fz4_CRD_. Second, the authors describe a putative glycosaminoglycan binding surface (surmised from the observed location of two SOS molecules in the crystal structure of the complex) composed of residues on both Norrin and Fz4_CRD_. Third, the authors point out a resemblance between the way in which Norrin interacts with Fz4_CRD_ and that in which Wnt8 interacts with Fz8_CRD_ (from a previous publication). Specifically, the position of the Wnt8 “index finger”, which interacts with the so called “site 2” on Fz8_CRD_, overlaps with strands b1 and b2 of Norrin, which interact with the analogous region on Fz4_CRD_. Furthermore, these beta-strands in Norrin show some structural equivalence with Wnt8. Finally, based on an analysis of solvent exposure, disease association, and lack of involvement in Fz4_CRD_ or GAG binding, the authors propose a positively charged, concave patch on the surface of Norrin as the putative binding site for Lrp5/6.

The core new result here is the crystal structure of the complex of Norrin with Fz4_CRD_/SOS and studies attempting to distinguish sites of GAG and LRP5/6 binding. Although the precise implications of these results for the signaling mechanism remain to be fleshed out, these results will form the basis of future studies and form an essential component of emerging models of Norrin/Wnt signaling.

There is a previously published study on the structure of Norrin fused to MBP (Ke et al. Genes and Development 2013), which closely resembles the structures of Norrin presented here (as the authors demonstrate in Figure 2—figure supplement 3, and Figure 4—figure supplement 4). Despite some overlap between the current manuscript and the previously published study, the paper under consideration represents an important contribution to the structural and biochemical analysis of a receptor-ligand interaction that plays an important role in both normal physiology and disease (and that is of considerable therapeutic interest given its potential involvement in vascular diseases of the retina).

The manuscript could be improved and shortened by more judicious citation of the literature and by clearing up—or acknowledging—weakness in the experiments and arguments, as indicated in the comments below.

Major issues to be addressed:

1) A principal scientific concern is the conclusion that the R121W site represents the LRP5/6 binding site. This conclusion may be correct, it just isn't fully supported by the data. Indeed, there is no direct measurement of the Norrin-Lrp5/6 interaction in the paper. Loss of signaling from the R121W mutation could also stem from loss of GAG interactions; this site is highly positively charged. The principal rationale for this site being the LRP5/6 interaction site seems to be that it is distinct from the better characterized (by the authors) GAG-binding site (SOS binds there and mutations knockout heparin binding), but it cannot be ruled out that the second site is also a GAG binding site. If the authors wish to make a claim for identification of an LRP binding site they should at least show that the R121W still binds a heparin column as tightly as wild type, or, better yet, show that R121W Norrin doesn't bind LRP-or tone down or qualify the section (already a bit qualified) on identification of the LRP binding site.

It is important that the authors address this issue, especially since it is one of the main points of discrepancy with Ke et al. (as the authors point out “Verification of GAG binding site”). The authors suggest that the site proposed by Ke el. al. to be the Lrp5/6 binding site on Norrin is in fact the SOS/GAG binding site and that their analysis has revealed the true Lrp5/6 binding site (notably, Ke et. al. did measure binding between Norrin and Lrp6).

2) [42] report the expression, crystallization, and X-ray structure determination of an MBP-Norrin fusion protein that is refolded after expression in *E. coli*. Therefore, claims of “technical difficulties generating recombinant Norrin”, in the present manuscript (Introduction), are overstated, if not misleading. The prior work was published more than a year ago, and the time has passed for the authors to present their work as competing with Ke, as opposed to building on Ke. The descriptions of the Norrin structure in the text and Figure 2 appear to be unnecessarily redundant with the earlier work, and the MS would benefit by focusing on comparisons of their structures with the earlier structure to highlight flexible/conserved regions, e.g. as shown in Figure 2—figure supplement 3.

3) The statement “…there have been no quantitative determinations of the binding affinity of Norrin for different CRD of Fz” (Results) is also misleading because both [42] and [89] report single-digit nanomolar affinities of Norrin for Fc-dimerized Fz4_CRD_. The affinity of the interaction between Norrin and Fz4_CRD_ is significantly different between this study (∼1 micromolar by SPR) and the prior study of Ke et. al. (∼10 nM by Alphascreen and the Octet Red instrument). These studies may be a bit different in that they use a dimerized CRD, which is presumably why the earlier authors measured tighter binding owing to avidity, but these measurements are certainly quantitative and even perhaps more physiologically relevant given the apparent Fz4 dimerization on cells. This should be noted and potential explanations for this discrepancy should be discussed.

4) The authors show that Norrin mutants in the SOS binding residues, especially the R107E/R109E/R115L mutant, show decreased activity in a luciferase assay for Wnt signaling (Figure 6) and fail to bind to heparin (Figure 6-figure supplement 2B). Can the authors provide any evidence that Glycosaminoglycans (GAGs) play a positive role in cellular responses to Norrin as the authors suggest and draw in all their models? The physiological relevance of this interaction for signaling is important to establish since it is a central aspect of the structure.

5) The authors propose that the SOS molecule in the structure (and presumably GAGs in cells; see model in Figure 6-figure supplement 2) bridges the interaction between Norrin and the Fz4_CRD_. As support, they reference a paper (Smallwood JBC 2007) that demonstrated heparin can enhance the affinity between Norrin and the Fz4_CRD_ by ∼10-fold. There is no direct experimental test of this model (except for a difficult-to-interpret experiment that excess SOS can decrease the signaling efficacy of Norrin). An easy (but important) test of this model would be to ask if (1) SOS or heparin can enhance the affinity of the Norrin-Fz4_CRD_ interaction in their SPR assay and (2) whether mutations in the putative SOS binding site of Norrin (e.g. the R107E/R109E/R115L triple mutation shown in Figure 6-figure supplement 2) can abrogate this effect of heparin/SOS (though they have little effect on the baseline Norrin-Fz4_CRD_ interaction).

6) The authors state that owing to the fact that FzCRDs are monomeric at high concentration in solution “CRDs of Fz receptors… are not involved in receptor dimerization.” Weak interactions in solution may yet be meaningful in 2D membranes and even if they don't contribute significantly energetically the observed contacts may form in Fz dimers (and influence the nature of the interaction) that form mostly based on favorable interactions of the TM regions.

7) It is curious that a single Fz4_CRD_ bound to the Norrin dimer in the absence of SOS. Do the authors have any rationalization for this unexpected stoichiometry (crystallization conditions, etc.)?

---

## [Author Response]

*Major issues to be addressed*:

*1) A principal scientific concern is the conclusion that the R121W site represents the LRP5/6 binding site. This conclusion may be correct, it just isn't fully supported by the data. Indeed, there is no direct measurement of the Norrin-Lrp5/6 interaction in the paper. Loss of signaling from the R121W mutation could also stem from loss of GAG interactions; this site is highly positively charged. The principal rationale for this site being the LRP5/6 interaction site seems to be that it is distinct from the better characterized (by the authors) GAG-binding site (SOS binds there and mutations knockout heparin binding), but it cannot be ruled out that the second site is also a GAG binding site. If the authors wish to make a claim for identification of an LRP binding site they should at least show that the R121W still binds a heparin column as tightly as wild type, or, better yet, show that R121W Norrin doesn't bind LRP-or tone down or qualify the section (already a bit qualified) on identification of the LRP binding site*.

It is important that the authors address this issue, especially since it is one of the main points of discrepancy with Ke et al. (as the authors point out “Verification of GAG binding site”). The authors suggest that the site proposed by Ke el. al. to be the Lrp5/6 binding site on Norrin is in fact the SOS/GAG binding site and that their analysis has revealed the true Lrp5/6 binding site (notably, Ke et. al. did measure binding between Norrin and Lrp6).

We thank the reviewers for urging us to give additional thought to these important issues. In response to their suggestions we have carried out a substantial amount of experimental work, the results of which we have added to the manuscript. We have also revised the text to clarify and discuss in detail points of agreement and discrepancy between our work and interpretation of results, and that of Ke et al. (42). We have paid particular attention, at all points in the manuscript, to check that our interpretations of the currently available data are clearly qualified when appropriate.

Firstly, we have carried out SPR binding experiments for the interaction of wild-type Norrin with Lrp6, and we have added these data to the manuscript (Figure 7). In our assay we measured an affinity of 2.87 μM for Norrin binding to Lrp6; Ke et al. reported a *K*_d_ value of 0.45 μM based on a homologous AlphaScreen competition assay using MBP-Norrin and Lrp6 (42). We note this difference in affinity values (presumably resulting from the very different assay formats) in the discussion.

Secondly, as suggested by the reviewers, we performed heparin binding assays. These experiments show Norrin R121W mutant still binds to heparin with a high affinity (Figure 8—figure supplement 1) similar to Norrin wild-type (Figure 7—figure supplement 1). However, as a result of performing the additional protein production and functional assays for the Norrin R121W mutant we discovered that this protein has lower solubility and stability than wild-type. We have added this information to the section on “Mapping a potential Lrp5/6 binding site on Norrin” and to the figure legend of Figure 8—figure supplement 1. This protein behaviour proved very problematic when we attempted to carry out SPR measurements of Norrin R121W mutant binding to Lrp6. We have therefore been unable to include this experiment in our revision.

Thirdly, we have toned down and qualified statements in the section “Mapping a potential Lrp5/6 binding site on Norrin”. We believe that the revised manuscript provides readers with a balanced assessment of the available data for the Norrin and Lrp5/6 interaction drawn from our current studies and those previously reported by Ke et al. Specifically, Ke et al. proposed the Lrp5/6 binding site comprises positively charged residues (Lys54, Arg107, Arg109, Arg115) and hydrophobic residues (Leu52 and Tyr53). They generated MBP-Norrin K54E/R109E mutant and found the double mutations impair Lrp6 binding in a competition assay (MBP-Norrin against the interaction of Lrp6_P1E1P2E2_ and DKK1 peptide). Actually, these results are in partial agreement with our proposed Lrp5/6 binding site, because our current studies also suggest that the Lys54 residue is involved in Lrp5/6 binding (Figure 8). However, our structural (Figure 5) and functional assays (Figure 7 and Figure 7—figure supplement 1) suggest that Arg109, the second residue of the MBP-Norrin K54E/R109E mutant, contributes to GAG binding rather than Lrp5/6 binding. We now clarify these points in the revised manuscript, placing our results in context with the previous studies with clear discussions in the sections on “Verification of GAG binding site” and “Mapping a potential Lrp5/6 binding site on Norrin”.

*2)*
[42]
*report the expression, crystallization, and X-ray structure determination of an MBP-Norrin fusion protein that is refolded after expression in* E. coli*. Therefore, claims of* “*technical difficulties generating recombinant Norrin*”*, in the present manuscript (Introduction), are overstated, if not misleading. The prior work was published more than a year ago, and the time has passed for the authors to present their work as competing with Ke, as opposed to building on Ke.. The descriptions of the Norrin structure in the text and*
Figure 2
*appear to be unnecessarily redundant with the earlier work, and the MS would benefit by focusing on comparisons of their structures with the earlier structure to highlight flexible/conserved regions, e.g. as shown in*
Figure 2—figure supplement 3*.*

We thank the reviewers for these suggestions. In response we have revised our text for Norrin protein production (Introduction section). We also have condensed our description of the Norrin structure (“Production of biologically active Norrin”), and rearranged Figure 2 and Figure 2—figure supplement 3 to address the reviewers’ comments and fully exploit the availability of the structure from the prior studies of [42]. However, we think our current shortened description of the Norrin apo structure is necessary to orient readers before the results sections on the complex and binding studies.

*3) The statement* “*…there have been no quantitative determinations of the binding affinity of Norrin for different CRD of Fz*” *(Results) is also misleading because both*
[42]
*and*
[89]
*report single-digit nanomolar affinities of Norrin for Fc-dimerized Fz4*_*CRD*_*. The affinity of the interaction between Norrin and Fz4*_*CRD*_
*is significantly different between this study (∼1 micromolar by SPR) and the prior study of Ke et. al. (∼10 nM by Alphascreen and the Octet Red instrument). These studies may be a bit different in that they use a dimerized CRD, which is presumably why the earlier authors measured tighter binding owing to avidity, but these measurements are certainly quantitative and even perhaps more physiologically relevant given the apparent Fz4 dimerization on cells. This should be noted and potential explanations for this discrepancy should be discussed*.

We appreciate the reviewers’ comments and have taken all of them into account in the revised manuscript. Specifically:

Ke et al., and Xu et al, indeed did quantitative measurements of Norrin binding to Fc-dimerized Fz4_CRD_ (89, 42), but they did not detect significant interactions, and therefore measure the binding affinity, of Norrin with the CRDs of other Frizzled receptors. From our structural comparison of Fz4_CRD_ with Fz8_CRD_ (Figure 9), we hypothesised that Fz8_CRD_ could have some ability to interact with Norrin. We therefore pursued studies to measure binding affinity values between Norrin and the CRDs of Frizzled receptors including Fz5_CRD_, Fz7_CRD_, and Fz8_CRD_. We agree with reviewers’ comment that our original wording is misleading and have changed the sentence “Because there have been no quantitative determinations of the binding affinity of Norrin for different CRD of Fz” to “To determine the binding affinity of Norrin for different CRD of Fz”.

Regarding the differences in affinity values from prior studies (89, 42), we originally briefly explained this issue in the figure legend of Figure 6—figure supplement 1. However, the reviewers made some excellent comments, which we have exploited in our revised text; we have moved our previous explanation from the figure legend of Figure 6—figure supplement 1 to the Discussion section and now discuss in more detail.

*4) The authors show that Norrin mutants in the SOS binding residues, especially the R107E/R109E/R115L mutant, show decreased activity in a luciferase assay for Wnt signaling (*Figure 6*) and fail to bind to heparin (Figure 6-figure supplement 2B). Can the authors provide any evidence that Glycosaminoglycans (GAGs) play a positive role in cellular responses to Norrin as the authors suggest and draw in all their models? The physiological relevance of this interaction for signaling is important to establish since it is a central aspect of the structure*.

We thank the reviewers for raising this point. We have indeed considered taking forward studies to explore the role of HSPGs in Norrin signalling. We agree with the reviewers that this is an interesting and important question to address, however, we consider that to do so fully will require substantial work that is beyond the scope of our current study. Therefore, in response to this comment:

Firstly, we have included new data from the SPR binding assay (Figure 7) to show the Norrin mutant (R107E/R109E/R115L) in the GAG binding site retains the ability to bind to Lrp6_P1E1P2E2_. We have made a new Figure 7, which is focused on the verification of the GAG binding site, and have added additional description of our results to the section on “Verification of GAG binding site”.

Secondly, we have added more description and clarified our discussion of HSPGs in the Discussion section.

Thirdly, we have taken note of the reviewers’ concerns (in comments 4 and 5) and on reflection have come to the conclusion that our schematics in various figures may give the reader a too simplistic impression of the possible ways in which GAGs may contribute to Norrin signalling. We have therefore decided to remove the old Figure 8 (cartoon model of Norrin induced receptor clustering and signallosome formation) and figure 6-figure supplement 2D.

*5) The authors propose that the SOS molecule in the structure (and presumably GAGs in cells; see model in Figure 6-figure supplement 2) bridges the interaction between Norrin and the Fz4*_*CRD*_*. As support, they reference a paper (Smallwood JBC 2007) that demonstrated heparin can enhance the affinity between Norrin and the Fz4*_*CRD*_
*by ∼10-fold. There is no direct experimental test of this model (except for a difficult-to-interpret experiment that excess SOS can decrease the signaling efficacy of Norrin). An easy (but important) test of this model would be to ask if (1) SOS or heparin can enhance the affinity of the Norrin-Fz4*_*CRD*_*interaction in their SPR assay and (2) whether mutations in the putative SOS binding site of Norrin (e.g. the R107E/R109E/R115L triple mutation shown in Figure 6-figure supplement 2) can abrogate this effect of heparin/SOS (though they have little effect on the baseline Norrin-Fz4*_*CRD*_
*interaction)*.

We appreciate the reviewers’ comments regarding the speculative nature of our model for the role of SOS/GAG molecule as an additional component of the Norrin–Fz4_CRD_ complex. We have undertaken a series of additional experiments and whilst we are able to provide substantial extra data in support of the distinct nature of the GAG binding site on Norrin we cannot provide definitive evidence for the particular importance of a bridging role in complex formation. We have accordingly re-balanced our discussion of this point throughout the revised manuscript. Specifically:

We apologize for not being clearer in our description of how excess SOS can inhibit Norrin signalling and have clarified our discussion of this experiment in the Discussion section by referencing two examples in which exogenous heparin and PG545 (a heparan sulphate mimetic) have been shown to inhibit Wnt signalling by competition with endogenous HSPGs for ligand and/or receptor interactions (4, 36). In an attempt to respond fully to the reviewers’ comments we have tried to measure the binding affinity of Norrin wild-type or R107E/R109E/R115L mutant with Fz4_CRD_ in the presence of SOS. However, we could not detect consistent, significant, changes in affinity using our SPR assay. On reflection we suspect that because the GAG binding site spans Norrin and Fz4_CRD_ (Figure 5), the inhibitory effects of separate Norrin-SOS binding and Fz4_CRD_–SOS binding events blocking formation of Norrin-SOS- Fz4_CRD_ (similar to in our SOS inhibition assay; Figure 7—figure supplement 1) may cancel out any stabilizing contribution to complex formation in these in vitro measurements.

It remains noteworthy that crystals of the native Norrin–Fz4_CRD_ complex only grew in the presence of SOS (Figure 4) and that the structural analysis reveals a SOS molecule spanning the interface between Norrin and Fz4_CRD_ (Figure 4). This ‘bridging’ action does suggest a mechanism by which to explain the observation of Smallwood et al. of an approximate 10-fold enhancement for the affinity between Norrin and Fz4_CRD_ in the presence of heparin. We therefore feel it is reasonable to retain some mention of the possible relevance of the ‘SOS bridge’ for promoting signalling complex formation in the revised manuscript.

In our original manuscript we reported that mutations (R107E/R109E/R115L) in the Norrin GAG binding site do not affect the ability of Norrin to bind Fz4_CRD_ (Figure 7) but do impair the activation of signalling (Figure 7). We have now carried out additional experiments to demonstrate Norrin R107E/R109E/R115L–Fz4_CRD_ complex formation (Figure 7—figure supplement 1), and interaction with Lrp6_P1E1P2E2_ (Figure 7) and have also confirmed that these mutations impair heparin binding (Figure 7). In combination, these data suggest that the GAG binding site is independent from the Fz4 and putative Lrp5/6 binding sites.

*6) The authors state that owing to the fact that FzCRDs are monomeric at high concentration in solution* “*CRDs of Fz receptors… are not involved in receptor dimerization.*” *Weak interactions in solution may yet be meaningful in 2D membranes and even if they don't contribute significantly energetically the observed contacts may form in Fz dimers (and influence the nature of the interaction) that form mostly based on favorable interactions of the TM regions*.

We agree with the reviewers. We cannot exclude the possibility that the CRDs of Fz receptors contribute to receptor dimerization in the cell membrane even though our *in vitro* solution studies indicate their interactions are very weak. We now state this when discussing these results.

*7) It is curious that a single Fz4*_*CRD*_
*bound to the Norrin dimer in the absence of SOS. Do the authors have any rationalization for this unexpected stoichiometry (crystallization conditions, etc*.*)?*

We think this is a very interesting question. Certainly such results are not unique, for example the case of colony stimulating factor-1 (CSF-1) and CSF-1 receptor, in which dimeric CSF-1 is bound to one molecule of CSF-1R in the crystal structure (Chen et al., 2008)

We had actually investigated what might be reason for our 2:1 complex structure, but decided to omit our findings from the original manuscript because we felt they might not be of sufficiently general interest. In our methylated Norrin–Fz4_CRD_ complex (Figure 4—figure supplement 1), we found that two lysine residues involved in the Norrin–Fz4_CRD_ interface are dimethylated in the uncomplexed subunit of the Norrin dimer. We have included this structural analysis in Figure 4—figure supplement 1 and added some comments to the section “The crystal structure of Norrin in complex with Fz4_CRD_”.